# UnidecNMR: automatic peak detection for NMR spectra in 1-4 dimensions

Charles Buchanan [1,2], Gogulan Karunanithy [1], Olga Tkachenko[1], Michael Barber[1], Michael T. Marty [1,3], Timothy J. Nott [4], Christina Redfield [4] & Andrew J. Baldwin [1,2] ✉

To extract information from NMR experiments, users need to identify the number of resonances in the spectrum, together with characteristic features such as chemical shifts and intensities. In many applications, particularly those involving biomolecules, this procedure is typically a manual and laborious process. While many algorithms are available to tackle this problem, their performance tends to be inferior to that of an experienced user. Here, we introduce UnidecNMR, which identifies resonances in NMR spectra using deconvolution. We demonstrate its favourable performance on 1 and 2D simulated spectra, strongly overlapped 1D spectra of oligosaccharides and 2D HSQC, 3D HNCO, 3D HNCA and 3/4D methyl-methyl NOE experimental spectra from a range of proteins. UnidecNMR outperforms a number of freely available algorithms and provides results comparable to those generated manually. Introducing additional restraints, such as a 2D peak list when analysing 3 and 4D data and incorporating reflection symmetry in NOE analysis further improves the results. UnidecNMR outputs a back-calculated spectrum and a peak list, both of which can be easily examined using the supplied GUI. The software allows interactive processing using nmrPipe, allowing users to go directly from raw data to processed spectra with picked peak lists.

Nuclear magnetic resonance spectroscopy is the most widely used experimental technique for characterising molecules, offering atomic resolution structural and dynamical information about chemical and biochemical systems. While NMR spectroscopy is ubiquitous, the analysis of NMR data is largely manual[1,2], which presents a substantial bottleneck. In related fields, for example, X-ray crystallography[3] and cryogenic electron microscopy[4,5], many viable automated analysis techniques have been devised, allowing them to become largely unsupervised high throughput methods[6]. This is not the case for NMR owing in large part to the specific challenges associated with identifying resonances in spectra[7]. We present here UnidecNMR, a general method for identifying resonances in NMR spectra, the quality of which is comparable to the results identified by experienced NMR

spectroscopists. We demonstrate the versatility of UnidecNMR through its application to a wide variety of spectra including small molecule 1D, 2D and 3D spectra of proteins, together with 3D and 4D NOE spectra, much of which is low signal to noise. We recently demonstrated the utility of UnidecNMR by implementing it into a full analysis pipeline for saturation transfer experiments, uSTA[8].

The first problem confronting an NMR spectroscopist once they have acquired and processed their data is to determine the number of resonances in their spectra, typically noting their chemical shifts and intensities. Three major issues render peak detection challenging in NMR: low signal-to-noise ratios, spectral overlap, and artefacts such as $T_1$ noise[9]. Many computational tools have been written to perform peak picking[10–17] although a 'standard' has yet to emerge that can

[1]Physical and Theoretical Chemistry, University of Oxford, Oxford, UK. [2]Kavli Institute for Nanoscience Discovery, Oxford, UK. [3]Department of Chemistry and Biochemistry, University of Arizona, Tucson, Arizona, USA. [4]Department of Biochemistry, University of Oxford, Oxford, UK. ✉e-mail: andrew.baldwin@chem.ox.ac.uk

operate in 1-4D. While peak-picking algorithms generally have outstanding performances on near-perfect data, their performance on 'real' problems tends to degrade, particularly in cases where signal-to-noise ratios start to approach 1, where resonances are heavily overlapped and where there are large variations in intensity due to their dynamics. Progress is largely made through manual analysis[18]. This unfortunately is not an effective option for either new students or for researchers unable to devote the necessary time for training and provides an argument for researchers to turn to other tools. We aim to address this here by providing a computational tool, UnidecNMR (Universal deconvolver for NMR). This software and associated graphical user interface (GUI) allow a user to process their multidimensional FIDs easily and interactively into spectra, execute our resonance identification algorithm and inspect the results, modifying where necessary. UnidecNMR can accelerate and simplify the workflow of both experienced and novice practitioners.

UnidecNMR is based on a Bayesian deconvolution algorithm that was previously developed for the analysis of Mass spectrometry data[19]. Deconvolution aims to separate 'sources' of resonances from acquired data using a given point spread function, analogous to how supra-resolution methods in light microscopy can locate light sources to better precision than the diffraction limit[20] (Fig. 1a). In this application, the point spread function is a peak shape function which needs to be effectively removed in order to locate the underlying resonances. To produce UnidecNMR, we optimised the naïve core of the mass-spec Unidec algorithm on a synthetic 1D dataset (Supplementary Fig. 4), before demonstrating successful application to a synthetic 2D dataset (Fig. 2).

When applied to experimental data, UnidecNMR is able to identify the individual multiplets of resonances in an extremely overlapped 1D NMR spectrum of a series of sugars (Figs. 1, 3). The algorithm is then tested against experimental data acquired on 5 uniformly labelled $^{13}C/^{15}H/^{1}H$ proteins of molecular weight ranging from 8.6 to 24.8 kDa dimer (2D $^{15}N$ HSQCs and 3D HNCO and HNCA spectra, Figs. 4, 5, Supplementary Table 1), and then on a 25.4 kDa 236-residue disordered protein where the spectra are highly crowded (Fig. 6). The testing dataset includes cases where all resonances are sharp and easily identified, to cases where signal to noise is low, and many resonances have low signal to noise because of exchange broadening. We finally analyse the performance of the algorithm on data acquired from 2 deuterated $^{13}CH_3$ ILV labelled proteins (3 and 4D methyl-NOESY spectra, Fig. 7).

We compare the peak-picking results from UnidecNMR to four frequently encountered algorithms that can be freely downloaded and already have user bases; PICKY, which relies on a singular value decomposition of spectra[10], WaVPeak, which takes advantage of a wavelet smoothing and clustering technique[11], NMRNet, which employs a convolutional neural network based on machine learning[12], and the intrinsic peak picker in Sparky[21], which in 2D (and 3D when used with a 2D peak list to restrict the search space) can detect local

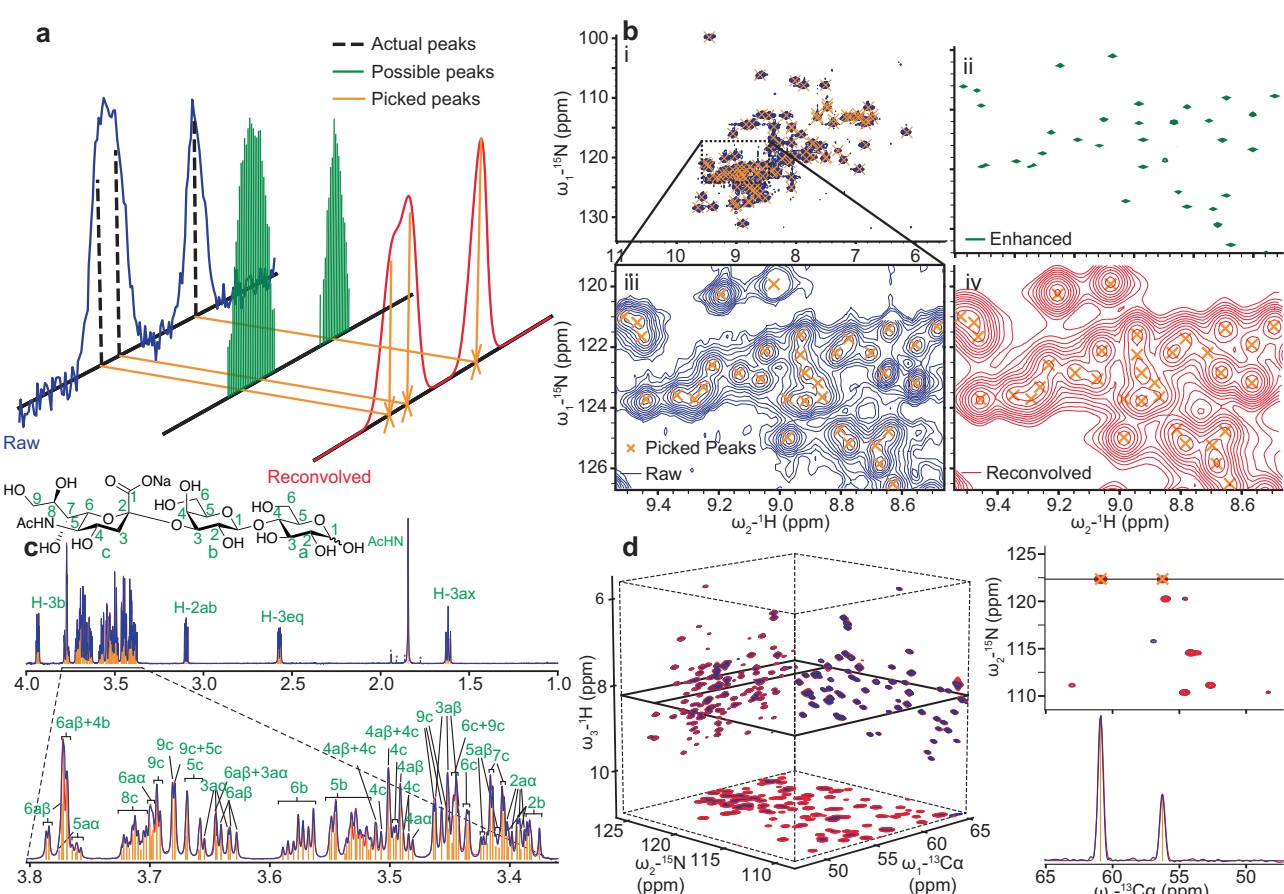

**Fig. 1 | UnidecNMR method and applications. a** A schematic of the UnidecNMR method: all locations above a noise threshold are initially possible peak locations before UnidecNMR iteratively filters them down to final locations. **b** UnidecNMR application to a 2D $^{15}N$-$^{1}H$SQC spectrum of αB-crystallin (i, blue). the reconvolved spectrum (iii, red) is easily compared against raw data (iv). Running UnidecNMR without the clustering results in a spectrum whose apparent sensitivity has been enhanced (ii, green). **c** Application to a 1D spectrum of 2,3-α-sialyllactose[8] showing a remarkable ability to discern individual assigned multiplets from a highly overlapped region of the spectrum. **d** Application to 3D HNCA spectrum of αB-crystallin, showing raw (blue) and reconvolved spectra in projection, and in 1 and 2D slices visualising how UnidecNMR simplifies resonance detection.

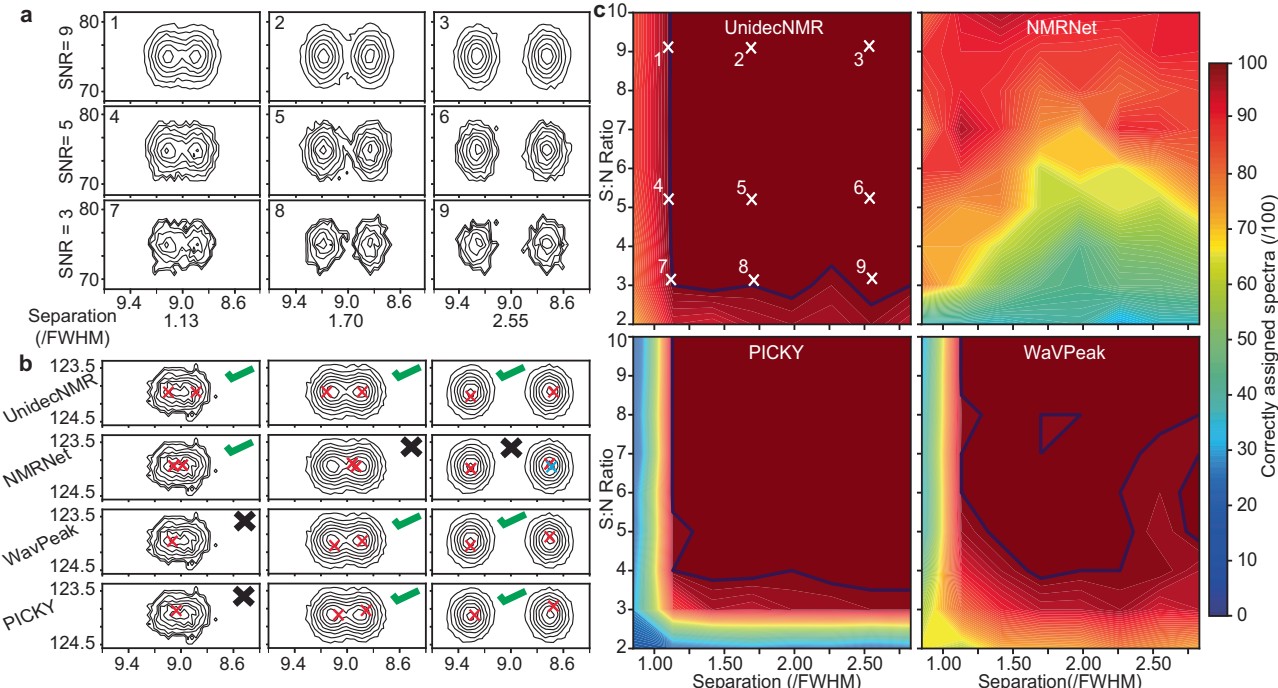

**Fig. 2 | Testing of peak picking algorithms on synthetic data.** Following optimisation of the UnidecNMR algorithm on 1D data (Supplementary Fig. 4) performance of a range of freely available peak pickers was assessed on simulated 2D data (7200 spectra). Two Gaussian resonances were simulated with a function of signal to noise (9 values ranging from 2 to 10) and separation (8 values, ranging from 0.85 to 2.6 in units of the full-width half maximum, FWHM of the simulated resonances). For each separation/SN combination, the seed for the random number that generates the noise was varied allowing us to derive an ensemble average over 100 repeats (see extended methods). In the UnidecNMR analysis, a Gaussian peak shape with the same width used for the simulation was used to deconvolve the data, with the one manually tuneable parameter, 'fac' set to 1.6. In the tests that follow, the width of the peak shape used by UnidecNMR was purposely mis-set for the purpose of testing which demonstrated that the results are reasonably agnostic of the precise value used, indicating that the peak shape chosen for analysis needs only to be 'roughly correct' (Supplementary Fig. 1). **a** Simulated data illustrate a range of signal-to-noise and separation values. The numbers 1–9 indicate where these example data were taken from the full dataset, c. **b** Illustrative examples of three spectra that provide a representative discrimination of the various algorithms tested that could be easily automated to iterate over the full dataset. Only NMRNet[12] and UnidecNMR were able to resolve the peak with the closest separation whereas only WaVPeak[11], PICKY[10] and UnidecNMR were able to distinguish the peaks with the widest separation. NMRNet overpicked one of the peaks on the third spectrum, which we have indicated in blue for clarity. **c** Overall performance on the entire simulated 2D dataset. Only UnidecNMR, PICKY[10] and WaVPeak[11] achieve 100% accuracy at the relatively trivial high separation/high signal-to-noise limit. The blue contour line represents the threshold above which 100% accuracy is achieved, allowing a convenient means by which to compare the different algorithms in this test. UnidecNMR outperforms the other algorithms on these synthetic datasets.

maxima. UnidecNMR demonstrated superior performance to the first three of these algorithms (we could not easily automate Sparky) on a synthetic 2D dataset (Fig. 2). On experimental data, UnidecNMR again substantially outperforms the other algorithms tested, and the resulting peak lists were either similar to those obtained by an experienced spectroscopist (Figs. 4–6 Supplementary Table 1), or in the case of the NOESY spectra, superior evidenced by the larger number of NOEs successfully identified that are consistent with the known structure (Fig. 7). We further demonstrate that supplying a 2D peak list when analysing 3 and 4D data (all 3 and 4D results in Figs. 4–7 use this approach) and including reflection symmetry when analysing NOE data can both substantially improve the results (Fig. 7). The algorithm is tested 'to destruction' and errors, both false positives and false negatives on real data have been individually characterised (Supplementary Figs. 5–7), each of which reflects ambiguous decisions on the edge of human judgement. The software is released with a GUI that facilitates rapid inspection and manual adjustment of the results (Supplementary Fig. 3). For example, false negatives can easily be identified within the GUI by overlaying the reconvolved and raw spectra (Supplementary Figs. 2, S7cii). The time of the calculation depends on the size and dimensionality of the data, but it completes in 5–30 s for a 2D spectrum, 30–120 s for a 3D, and several minutes for a 4D on a 2021 MacBook Pro equipped with an M1 Pro processor and 16GB of RAM. The package has been tested on Windows, Mac and Linux environments. Overall, our algorithm is a tool for the analysis of

1-4D NMR data that is free for academic use, that at the very least, provides an excellent 'starting point' for both new and experienced users to facilitate rapid and effective analysis of NMR data.

## Results
### Theory
The kernel of the algorithm was originally developed to analyse mass spectrometry data[19], a problem that shares many features with NMR data analysis. The method relies on the assumption that a spectrum of intensities, $I$, can be reasonably expressed as a convolution of a peak shape function, $g$, and an array of delta functions or sources, $f$, each of which spans the same set of spectral frequencies, $i$. The algorithm aims to perform the deconvolution that removes the peak shape function from the data, providing a user with a list of sources, $f$, that dictates both the peak positions and intensities. The kernel iterates ($t$) on the intensities of each spectral element:

$$f_i^{t+1} = f_i^t \frac{I_i}{(f^t * g)_i} \qquad (1)$$

When the back-calculated spectrum, $(f * g)$, has the same intensity as the data $I$ at frequency $i$, then the ratio is equal to 1, $f_i^{t+1} = f_i^t$ and the algorithm has converged[19]. The action of the algorithm is to suppress sources that are adjacent to the true 'centre' of a resonance. A user supplies a noise threshold and a peak shape function $g$ (Fig. 1a). In our

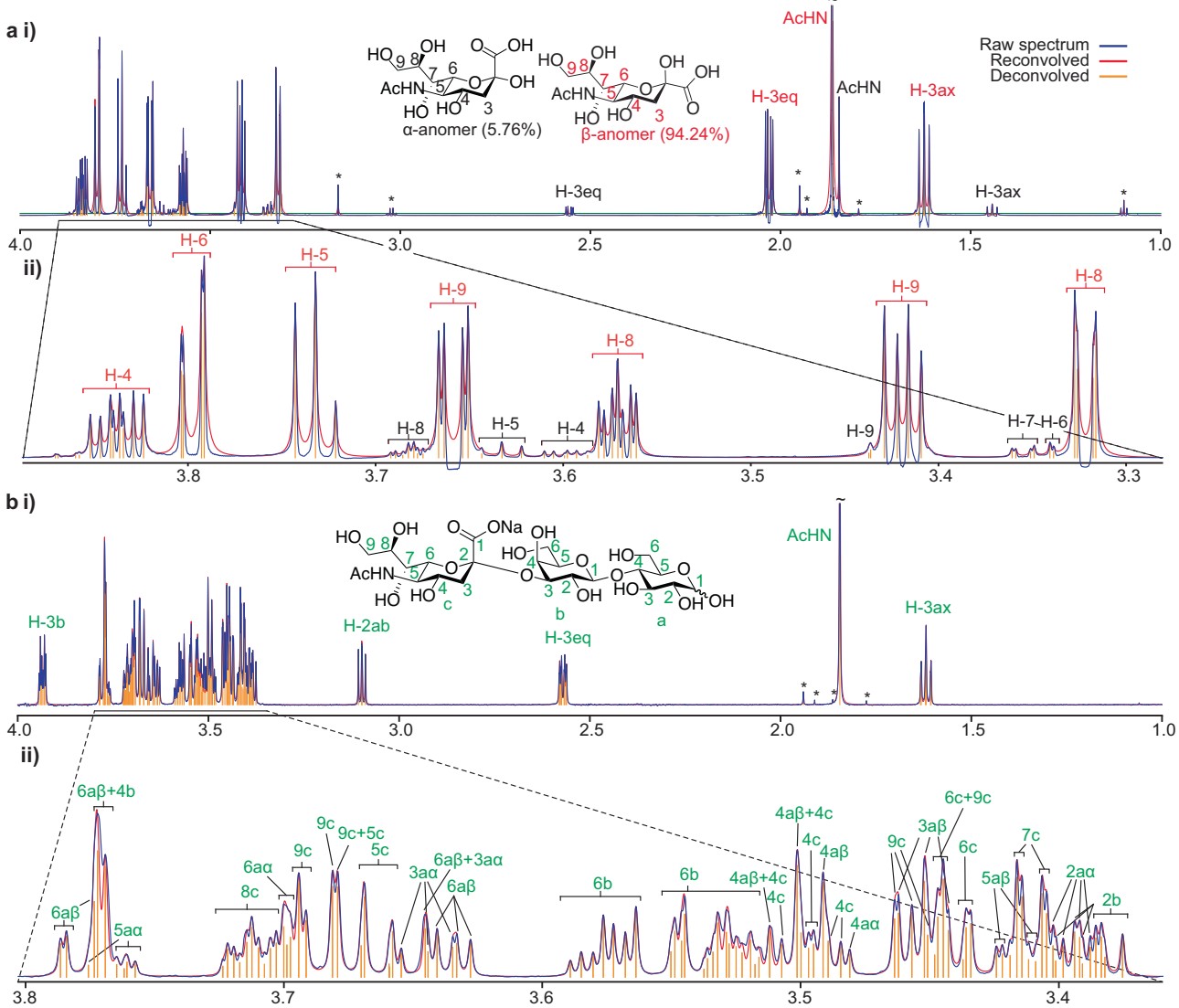

**Fig. 3 | Performance of UnidecNMR on 1D spectra.** 1D proton spectra (blue) of N-Acetylneuraminic acid (**a**) and 2,3-α-sialyllactose (**b**) together with the reconvolved spectrum from UnidecNMR (red) as shown previously[8]. The deconvolution was performed with the tuneable parameter fac = 1.4 using a peak shape fitted on isolated resonances using the GUI. In each case, the multiplets identified by UnidecNMR (yellow) can be mapped to known assignments obtained from standard multi-dimensional methods[8]. The peak shape used for the analysis was obtained by fitting the most intense peak using the UnidecNMR GUI. There are small variations in peak shape over the spectrum (see e.g., peak 6c at 3.45ppm in **b**), but this does not compromise performance. **a** Two interconverting anomeric forms, α and β, are observable. UnidecNMR functions well even when evolution due to J coupling is present in the spectrum, imposed by the delays associated with using excitation sculpted water suppression. **b** Two interconverting anomeric forms associated with sugar a are observable (inset). Unique assignment cannot be obtained from a single 1D NMR spectrum, but each peak identified by UnidecNMR corresponds to assignments achieved using standard methods[8], as indicated (green). No water suppression was used for this spectrum and so there are no distortions due to J coupling present, which will yield optimal UnidecNMR performance.

---

implementation, the initial intensities are set to the initial intensities of the raw data found at each position, and then iteratively adjusted according to Eq. (1) until convergence, where the final values in $f$ provide the central locations and intensities of the picked peaks. To determine convergence, the changes made in intensities are assessed in each step, and when these fall below a user-specified threshold, or when the calculation exceeds a pre-specified number of maximum iterations, the calculation stops. The selected peak shape $g$ should be a reasonable match for the average resonance in a spectrum, although the final results are reasonably tolerant to this parameter being mis-set (Supplementary Fig. 1a), as expected in the case of experimental data where there is a wide range of peak shapes. The algorithm can work on data of arbitrary dimensionality, which renders it highly amenable to NMR analysis.

### Performance on synthetic data

To test the algorithm, a dataset of 40,500 1D NMR spectra was simulated by placing two resonances at predefined separations, and introducing noise sampled from a normal distribution to obtain a predefined signal-to-noise ratio (Supplementary Fig. 4a, b, e). Naïve implementations of Eq. (1) showed promise for peak detection but tended to 'over-pick' the spectrum (Supplementary Fig. 4a, c). This result follows from there being no unique solution when fitting an arbitrary number of Gaussian functions to a Gaussian function[22], and so the final arrangement of peak locations, $f$, themselves, tend to resemble a Gaussian function. From the perspective of peak detection, this amounts to a failure, as too many resonances are 'picked'. This does, however, provide an unexpected feature of this algorithm—one can apply it 'naïvely' with a peak shape set to be deliberately too wide

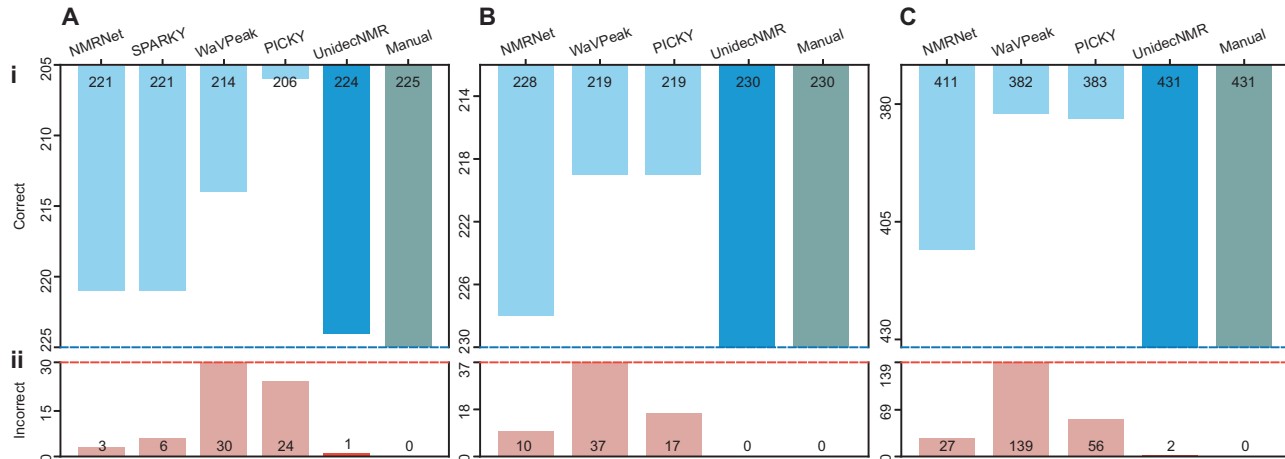

**Fig. 4 | Relative performance of UnidecNMR versus a series of alternative peak-picking algorithms on 'real' NMR data acquired on 4 different proteins.** Counts of correct (i) and incorrect picks (ii) are shown for a set of $4 \times 2D$ $^{15}N^1H$ HSQCs (**A**), 3D HNCO (**B**) and 3D HNCA (**C**) spectra as described in the text. The results were scored against independently determined peak lists by a skilled user. A detailed tabulation of these results and scoring criteria is provided (Supplementary Table 1). When running UnidecNMR in 2D, the tuneable parameter 'fac' was set to 1.4, and in 3D, 1.6. In all cases, the peak shape parameters were fitted on isolated resonances using the GUI, and when running in 3D, a 2D peak list was provided ('boring' mode) as described in the text.

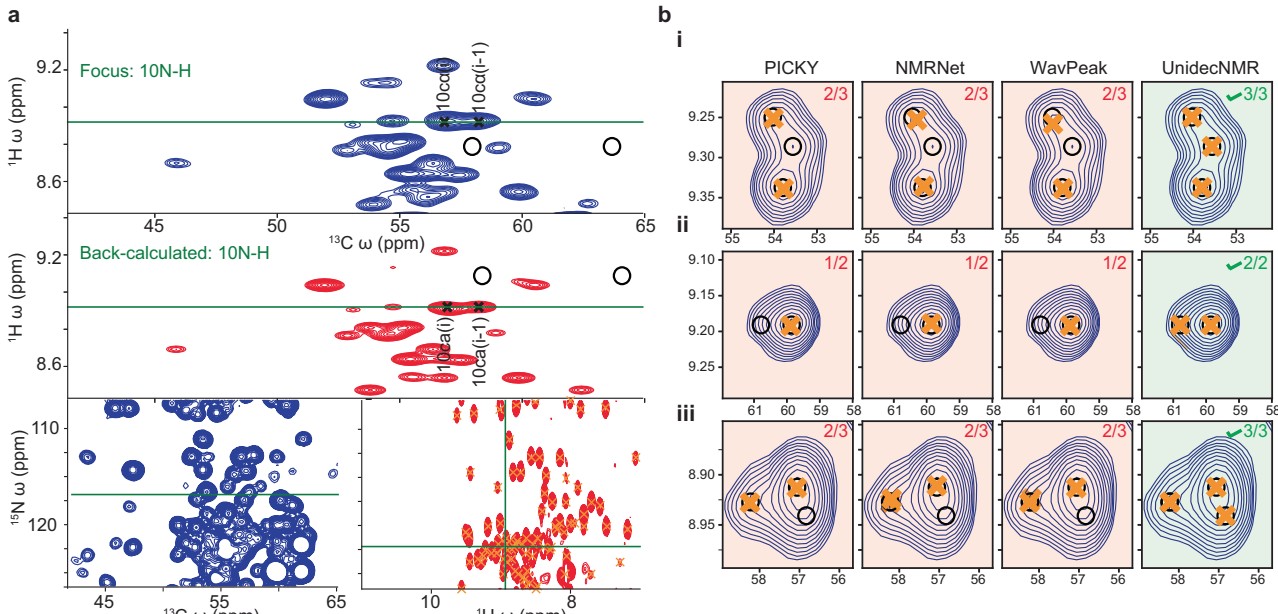

**Fig. 5 | Analysis of 3D HNCA of αB-crystallin using UnidecNMR and representative examples of errors obtained using the various algorithms. a** After processing, the slice for each peak can be selected to inspect the peak picks and to compare the raw data to the back-calculated spectrum. The location of the slice in the 3D is indicated, with respect to the overall projections. **b** Representative errors from the various peak pickers[10–12,21]. (i) In a 3D HNCA spectrum, three highly overlapped resonances can be seen. UnidecNMR could identify this while the other algorithms failed. (ii) In a 3D HNCA, the overlap between these two weak peaks is suggested by the asymmetry. Only UnidecNMR correctly identified the underlying resonances. (iii) WaVPeak[11] and PICKY[10] were unable to pick relatively weak resonances defined by relatively few data points, as exemplified by this HNCA.

and obtain a new NMR spectrum with substantially enhanced apparent resolution (Fig. 1bii).

After some development, the final version of UnidecNMR runs in stages. Equation (1) is initially executed with a peak shape function $g$ whose FWHM is artificially increased by a factor 'fac'. This suppresses the tendency of the algorithm to pick peaks remote from the true centre (Supplementary Fig. 4a, b). Second, we implemented a clustering algorithm that combines intensity within a predefined window whose width is characterised by the parameter 'squash'. Both are specified as a multiple of the FWHM of the input peak shape. Running the algorithm against our database of simulated data allowed values of *fac* and *squash* to be optimised in order to produce the maximum number of correct results (Supplementary Fig. 4e, f). In this test, a broad range of parameter space was identified where 100% accuracy was achieved. Within the final implementation, 'squash' was fixed to 0.725/FWHM and is determined automatically from the user-supplied peak width, and 'fac' is the only user-supplied parameter. In practical implementations, this can be adjusted typically with the range 1.4–1.6, with a value of 1.4 being suitable for 1D/2D data and 1.6 better suited for 3D/4D (see also 'Standard settings for using UnidecNMR' in

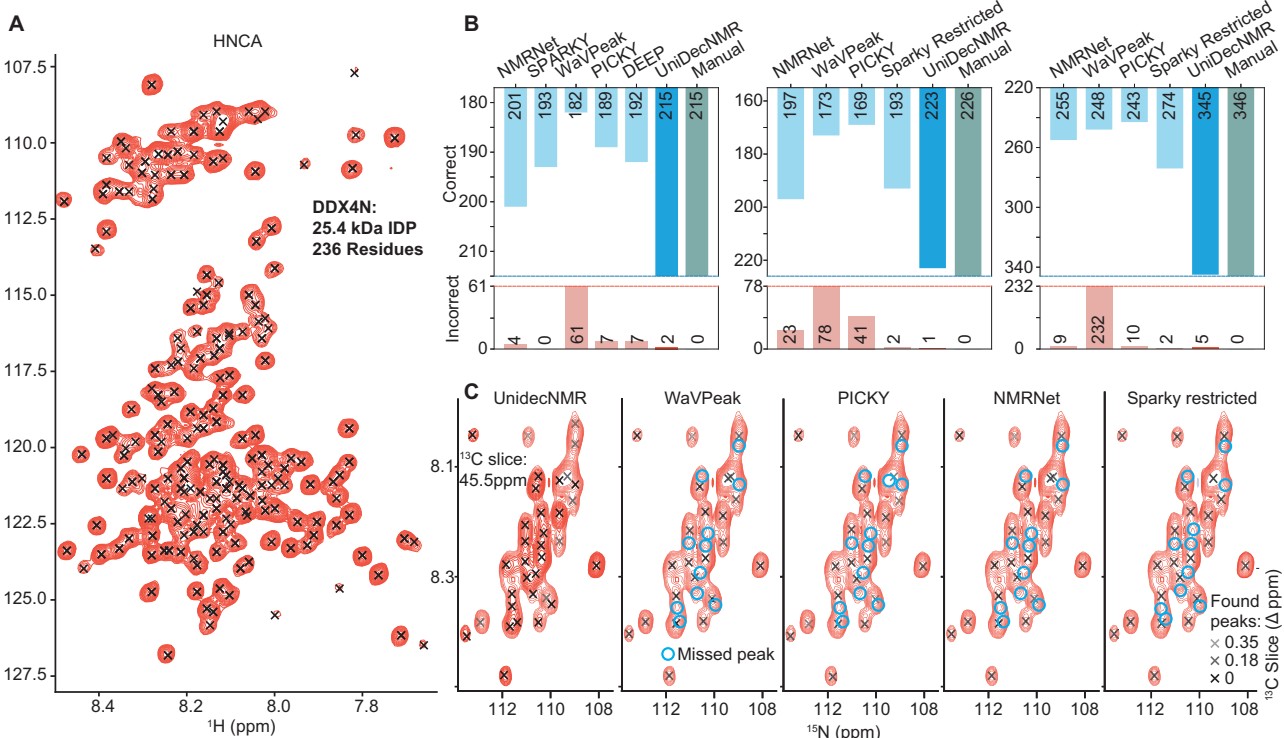

**Fig. 6 | Comparison of different peak-picking methods on the 236-residue intrinsically disordered protein DDX4N1. A** An NH projection of an HNCA acquired on a 750 MHz spectrometer, superimposed on a peak list derived from UnidecNMR results from a high-resolution HSQC (Supplementary Fig. S6B)[40]. As expected for a large IDP, the centre of the spectrum is heavily overlapped. **B** An indication of the false negatives and false positives of the different methods tested on a 2D $^{15}$N HSQC, and 3D HNCO and HNCA spectra. (i) On a high-resolution 2D $^{15}$N HSQC, similar performance was found for Sparky, WaVPeak, PICKY, NMRNet, and DEEP picker with ca. 30 false negatives and between 0 and 66 false positives. By contrast, UnidecNMR picked exactly the same resonances as an experienced user with two false positives (Supplementary Fig. S7A). For 2D analysis, the UnidecNMR tuneable parameter 'fac' was set to 1.4. (ii/ii) Similar performance was found in 3

dimensions, with UnidecNMR picking over 100 more resonances in the two spectra that were missed by the other methods. The tuneable parameter 'fac' was set to 1.6 for 3D data. The specific peaks that were missed by UnidecNMR are analysed (Supplementary Fig. 7), arising from resonances appearing in the spectrum that were not in the HSQC, arising most likely due to some sample degradation. **C** A slice from a heavily overlapped region in the HNCA illustrates the performance of UnidecNMR and the places where the other algorithms fail to spot resonances. Overall, the performance of UnidecNMR is almost identical to those obtained by an experienced user. As these are N-H slices through a 3D spectrum, the distance of an identified peak from the current, carbon, slice is indicated with the colour of the peak, as shown in the key (bottom right).

Supplementary). These values effectively set a limit on the resolving power of our algorithm yet still allow asymmetric 'shoulders' to be readily identified.

With optimised values of 'fac' and 'squash', the performance of UnidecNMR was effectively perfect, provided that the signal-to-noise ratio is greater than 3 and the separation of resonances is greater than 1.13 FWHM (Supplementary Fig. 4d for 1D, Fig. 2c for 2D). This is a physically reasonable outcome, as these limits approximately coincide with limits where a user would be confident in visually identifying two resonances. Below these thresholds, the success of the algorithm falls away from 100%, and its success depends on the exact shape of the noise profile in an individual spectrum. It nevertheless remains reasonably successful outside these windows and typically fails in cases where an experienced user would also be unconfident.

We next performed a similar test on 7200 simulated 2D spectra using the optimised parameters (Fig. 2). Here, it was possible to compare results from UnidecNMR to the other freely downloadable algorithms. The success of UnidecNMR in 2D was very similar to its performance in 1D and was more effective than the other methods tested (Fig. 2c). PICKY and WaVPeak both showed excellent performance, but require higher S/N, and larger peak separations than UnidecNMR to obtain a 100% success rate. Notably NMRNet performed very poorly in this test (Fig. 2c), although we note this program performed substantially better when tested against 'real' experimental data recorded on protein samples (Fig. 4).

To quantify the accuracy of the respective algorithm's identified peak locations, we calculated the difference between the found and known locations in 'correct' spectra (Supplementary Fig. 8). This shows that UnidecNMR performs better than all other algorithms at higher signal-to-noise ratios and lower separations, a problem typically faced in experimental data (i.e., heavy peak overlap where signal to noise is reasonably high). Further, we quantified the reliability of intensities extracted by UnidecNMR and found that except at very low separations, the error was <10% (Supplementary Fig. 9).

To run UnidecNMR in general, both a noise threshold and parameters that describe the peak shape in all dimensions must be supplied. For convenience, we have implemented a general pseudo-Voigt function that describes a mixed Gaussian/Lorentzian function. This is parametrised by Lorentzian ($\sigma_L$) and Gaussian ($\sigma_G$) FWHM values in ppm, the Euclidean distance (x) from the centre of the peak in ppm and a mole fraction $n$ that determines the degree of Gaussian ($n = 0$) and Lorentzian ($n = 1$) character.

$$P_i(x, n, \sigma_L, \sigma_G) = (1 - n) \exp\left(-\frac{x^2 \sqrt{2\ln 2}}{\sigma_G^2}\right) + n \left(\frac{\left(\frac{\sigma_L}{2}\right)^2}{x^2 + \left(\frac{\sigma_L}{2}\right)^2}\right) \quad (2)$$

Optimal performance is obtained when the shape characteristics either match or are slightly narrower than those observed although the algorithm is remarkably tolerant to mis-setting the peak shape (Supplementary Figs. 1a, 4f). In practical applications, the algorithm can be

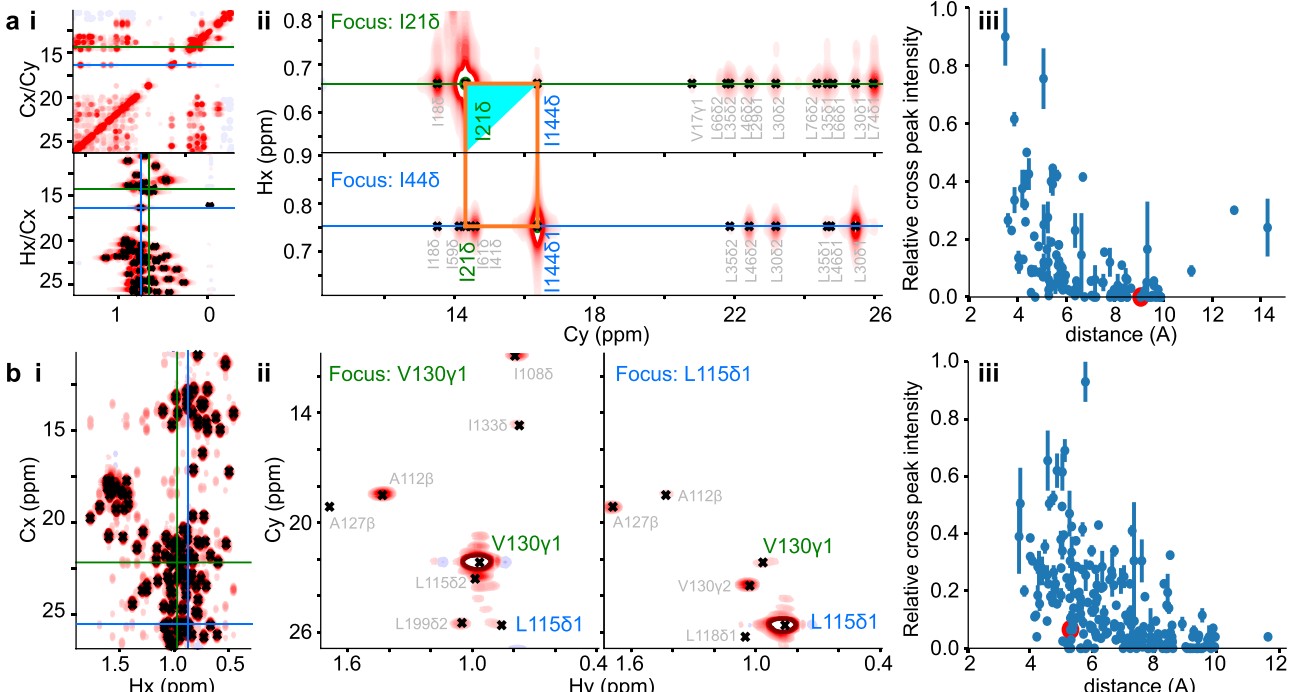

**Fig. 7 | Automatic picking of 3 and 4D methyl NOE spectra.** Results from a 3D methyl NOESY spectrum from ATCase (**a**) and a 4D methyl NOESY spectrum of EIN (**b**). These figures were generated from outputs within our GUI. (i) The location of two selected slices is indicated with respect to the relevant projections. It is desirable when analysing 4D spectra to take 2D slices that have lower resolution than the reference planes that include the direct dimension, as shown for EIN. (ii) The corresponding slices focusing on identified NOEs from a pair of resonances, and a cross peak between them. The specific cross peak feature is indicated (orange). Orthogonal views can be selected within the GUI allowing a user to verify that all resonances are centred on the selected plane. The deconvolved version of the spectrum can also be shown side-by-side with the raw data. (iii) The cross peak signal intensity, shown as a mean and standard deviation of the two reciprocal resonances, shown versus the expected C-C distance from the corresponding structure. 460 pairs of cross-peaks were identified for ATCase and 660 for EIN, overall 60% more than obtained from a manual analysis[29] (294, 420 respectively). The picked peaks fall within a sensible range of distances indicating that the NOEs are consistent with the expected structures (iii).

run iteratively and compared to the raw data to manually determine appropriate peak shape parameters. Within our GUI, it is also possible to select several intense, isolated resonances in a spectrum and run a conventional optimiser to obtain a reasonable estimate for parameters to describe the peak shape (Supplementary Fig. 3).

**Performance against experimental data**
Having demonstrated that our algorithm can function well against synthetic data, we tested it against experimental data. Initially, we ran the algorithm on 1D NMR spectra of sugar molecules that contain a very large number of highly overlapped resonances. The algorithm was able to identify individual multiplets even in heavily overlapped regions (Fig. 3), each of which was consistent with the known assignment obtained using conventional methods.

To test further, 2D $^{15}$N HSQC, and 3D HNCO and HNCA spectra from three proteins, HSP27[1,23], ubiquitin[24] and αB-crystallin[25] (NMR data unpublished) were analysed. Spectra from these proteins provide a range of difficulties, ranging from Ubiquitin and HSP27, where the resonances are sharp and well resolved, to αB-crystallin, where there are substantial contributions from chemical exchange leading to a range in peak intensity and shape[25]. The performance of the algorithm was measured against a peak list determined manually.

When assigning the backbone of a protein, 3D spectra such as the HNCO and HNCA are analysed simultaneously[26]. Mirroring this, a 2D H/N peak list was prepared using only the $^{15}$N HSQC and HNCO results and supplied to the algorithm as a restraint for the starting locations for peak positions. The resulting 'boring' mode produced excellent results. We provide this as an option in the software and recommend using this when analysing 3 and 4D data. In this vein, the HNCO and NHSQC spectra were analysed in isolation while we employed a 2D

peak list for the HNCA (Fig. 4). As with the synthetic data, UnidecNMR substantially outperformed the other algorithms, achieving almost 100% success rates, identifying all peaks and missing only 1 resonance (Fig. 4, Supplementary Table 1). This error deserves some attention. In fact, we would not expect this missing resonance to be identified by a user given only this spectrum (Supplementary Fig. 5), due to overlap, though its presence is confirmed by additional 3D experiments. NMRNet, however, did identify two resonances in this location, and so we report it as a peak missed by UnidecNMR. We note that in this test, NMRNet tended to over-pick NMR spectra (Fig. 4). As our algorithm provides a back-calculated spectrum, it is straightforward to compare its results to the raw data within our GUI (Supplementary Fig. 3).

One spectrum of note was an HNCO acquired on HSP16.5 (unpublished data), where the decoupler was mis-set, resulting in a triplet for each peak in the $^{13}$C dimension (Supplementary Fig. 1b). By increasing the effective peak width used for calculations, we were able to account for this deficiency. By contrast, the other algorithms found analysing this specific experiment highly challenging.

To increase the challenge further, we then analysed $^{15}$N HSQC, HNCO/HNCA assignment spectra from a 236-residue intrinsically disordered protein DDX4N1 (Fig. 6). Disordered proteins are relatively challenging targets because the range of chemical shifts spanned is much narrower than for a folded protein and so overlap of resonances is substantially higher. As before, UnidecNMR outperformed the other algorithms and returned results comparable to those obtained from an entirely manual analysis. The small number of specific errors where UnidecNMR diverged from the manual peak picks were analysed (Supplementary Figs. 6, 7) were classified as ambiguous in the human derived assignment.

Finally, we sought to test the algorithm against methyl-NOE data. Two spectra were analysed; a 3D dataset from a dimer of regulatory chains of aspartate transcarbamoylase from *E. coli* ATCase, acquired in the laboratory of Prof. Lewis Kay[27] and a 4D spectrum from the N-terminal domain of *E. coli* Enzyme I (EIN1), acquired in the laboratory of Prof. Marius Clore[28]. In both cases, a 2D H/C peak list was initially prepared from a high-resolution 2D HMQC spectrum. As the proteins are deuterated with only $^{13}CH_3$ methyl groups labelled, the expected distance of NOEs will be substantially longer than those spanned in uniformly labelled samples, frequently extending to a C-C distance of 10 Å[29]. Moreover, cross-peaks are expected to be symmetric with minimal spin diffusion, such that if A->B is present then we also expect B->A[30]. We provide an option in UnidecNMR to impose this requirement on the spectrum and to seek out only cross-peaks related by reflection symmetry. The resulting peak list is returned as a set of correlations between two resonances in the original 2D peak list, ready for use in assignment software or in structure calculations.

For ATCase/EIN, 460/660 cross-peaks were picked by UnidecNMR, an increase of 60% from those picked previously by an experienced user whose results were used for assignment (294/420)[29]. The intensity of the resonances was plotted against the known distance spanned in the protein to determine if the picked NOEs are consistent with the expected structural forms (1d09[31]/1eza[32] ATCase/EIN) (Fig. 6A). Both plots revealed the vast majority of the NOEs to be within 10 Å and so are consistent with the known structures (Fig. 6A) and have a similar pattern to those picked manually[29]. For ATCase, several relatively intense resonances were identified both manually and by UnidecNMR that would indicate a distance >10 Å[29]. As noted previously[27], these indicate a fluctuation in the loop adjacent to leucine 7 of ATCase present in the truncated dimer that is not present in the WT structure used for analysis. In the case of methyl NOE spectra, as 60% more cross-peaks were identified, UnidecNMR performance exceeds that of an experienced user.

## Discussion

When publishing the first 'triple resonance' $^{13}C/^{15}N/^1H$ backbone assignment experiment, the HNCO, the authors noted that "because of the low level of resonance overlap in the 3D spectra, much of the 3D peak picking can be done in a fully automated manner."[33]. While this experiment remains widely used, the goal of full automation has yet to be realised.

To move further towards this goal, UnidecNMR provides a powerful computational tool for picking resonances in 1-4D NMR spectra. In practice, a user supplies parameters that describe a peak shape that will be ultimately used to back-calculate the spectrum, together with a noise threshold. Both can be either estimated by the software or manually adjusted in an iterative manner by a user. The target spectrum and the back calculation are in nmrPipe[34] format. The resulting picked peaks and back-calculated spectra can be inspected either in our GUI or in any other preferred visualisation software. Against both synthetic and experimental data, the performance of UnidecNMR is substantially improved over the alternative freely downloadable peak-picking algorithms tested in this article.

The algorithm offers two additional modes that can use prior knowledge to improve performance, neither of which is typically possible with existing peak-picking software. Providing a 2D peak list limits the search space substantially in 3 and 4D applications, and in the case of 3 and 4D methyl NOE spectra, a user can further specify the additional requirement that symmetry will be imposed on any cross-peaks. These peak lists can then be used for assignment[29] or passed to structural calculations.

The GUI is written in wx-python and uses the Python package matplotlib[35] to generate figures (Figs. 5a, 6, Supplementary Figs. 2, 3). Matplotlib is widely used by NMR users to visualise data[35] and our software makes it easy for the plotting functions to be manually edited to cater for individual preferences. We are taking advantage of the

outstanding nmrGlue package[36], which allows NMR spectra to be read into Python. The deconvolution software is written in C++ which can be executed from the command line and so can be incorporated into automated workflows outside of our own GUI. The GUI and deconvolution program have been tested in Linux, Mac and Windows environments. The simulated data comprising 40,500 1D and 7200 2D simulated spectra (Supplementary Figs. 2, 4) and the experimental data (Supplementary Table 1) will be made freely downloadable to enable rapid and systematic comparison of peak-picking algorithms going forward.

Overall, against the experimental data tested here, we find the performance of UnidecNMR to be either comparable or, in the case of methyl NOE data, substantially superior to the results generated by an experienced spectroscopist (Pritišanac et al.[29]). A GUI is provided to enable a user to quickly screen through the picks to both check the results and manually amend, as required. The GUI also generates and executes nmrPipe[34] and SMILE[37] scripts for interactive processing of 2-4D NMR. This allows a user to go straight from FIDs to processed and picked spectra within one software environment using a few 'clicks'. While it is possible to add the algorithm to a fully automatic pipeline, we nevertheless recommend inspecting the results manually. Either way, by substantially reducing the time taken to analyse 3 and 4D spectra, UnidecNMR has promise to both accelerate the workflow of spectroscopists and reduce the barriers for non-specialist laboratories to undertake a biomolecular NMR analysis to address their research questions. The software and benchmark are free for academic use and can be downloaded from http://UnidecNMR.chem.ox.ac.uk.

## Methods

### Simulation methods

NMR spectra with 2 peaks were simulated in 1 (Supplementary Fig. 4) or 2 dimensions (Fig. 2) using a Gaussian function with a predefined width. The difficulty of each spectrum was controlled by two axes: the separation of the two peaks and the signal-to-noise ratio (*y*-axis, 1D−Supplementary Fig. 4c, d, f, 2D−Fig. 2c, varied between 2 and 10 in both cases).

For 1D spectra, a peak location function of 400 frequency points was defined as zero everywhere except two points separated by the given separation value (expressed as a function of FWHM, *x*-axis 1D−Supplementary Fig. 4c, d, f, 2D−Fig. 2c ranging from 0.85 to 2.6 in both cases). A standard Gaussian peak shape function was defined and convolved with this peak location to give a noiseless spectrum.

To define the signal to noise for each spectrum, a random noise function was produced by drawing from a normal distribution. To mirror what is typically done in the analysis of experimental spectra, the highest intensity/standard deviation was taken from a defined region of this signal-less function. The magnitude of the noise function was then adjusted before being added to the signal to produce a final spectrum with the required signal-to-noise ratio.

500 1D spectra were simulated for each of 9 signal-to-noise values and 9 inter-peak separations (Supplementary Fig. 4). The same protocol was repeated for 2D spectra, and we chose 100 spectra for each of the 9 signal-to-noise ratios and 8 separations (Fig. 2). As we were producing so many spectra, care was taken to vary the random seed to ensure the noise distribution was not being repeated.

### Experimental methods

All data, processing scripts and analysis are included as part of a benchmark available for download as described in the data and code availability statement. Methods to produce the materials described and data acquisition are described below.

***αB-crystallin*** (86 residues, present as a dimer, 19.8 kDa):
Sequence:
MRLEKDRFSVNLDVKHFSPEELKVKVLGDVIEVHGKHEERQDEHGFI SREFHRKYRIPADVDPLTITSSLSSDGVLTVNGPRKQVS

Residues 68–153 of Human αB-crystallin are known to form a predominant dimer (19.8 kDa) at equilibrium, although all spectra show signs of exchange broadening, consistent with previously observed monomer-dimer interchange[25,38]. All spectra of αB-crystallin core domain were recorded at 288 K on an Oxford Instruments 600 MHz spectrometer equipped with a Varian Inova console and a 5 mm triple resonance room temperature probe with z-axis gradients.

The 2D $^{15}$N-$^{1}$H sensitivity-enhanced HSQC was recorded with 100 ($^{15}$N) and 1024 ($^{1}$H) complex points and 2110 Hz ($^{15}$N) and 9000 Hz ($^{1}$H) sweep widths using an interscan delay of 1 s and 16 scans per FID for a total duration of 60 min. The sensitivity-enhanced HNCO was recorded with 40 ($^{13}$C), 15 ($^{15}$N) and 1298 ($^{1}$H) complex points with 1400 Hz ($^{13}$C), 1400 Hz ($^{15}$N) and 9000 Hz ($^{1}$H) sweep widths using an interscan delay of 1.5 s and 8 scans per FID for a total duration of 8 h 52 min. The sensitivity-enhanced HNCA was recorded with 30 ($^{13}$C), 15 ($^{15}$N) and 1152 ($^{1}$H) complex points with 4500 Hz ($^{13}$C), 1210 Hz ($^{15}$N) and 9000 Hz ($^{1}$H) using an interscan delay of 1.2 s and 16 scans per FID for a total duration of 10 h 43 min.

***HSP27core*** (88 residues, present as a dimer 19.8 kDa)[1,23]:
Sequence:
GVSEIRHTADRWRVSLDVNHFAPDELTVKTKDGVVEITGKHEERQ
DEHGYISRCFTRKYTLPPGVDPTQVSSSLSPEGTLTVEAPMPK

Residues 86–171 (a.k.a. the core domain) of Human HSP27 are known to form a predominant dimer (19.8 kDa) at equilibrium, although all spectra show signs of exchange broadening, consistent with previously observed monomer-dimer interchange. The 2D $^{15}$N-$^{1}$H sensitivity-enhanced HSQC was recorded at 298 K on an Oxford Instruments 750 MHz spectrometer equipped with a Bruker Avance III HD console and a 5 mm TCI cryoprobe with z-axis gradients.

128 ($^{15}$N) and 1024 ($^{1}$H) complex points were acquired with 2257 Hz ($^{15}$N) 10,000 Hz ($^{1}$H) sweep widths using an interscan delay of 1.4 s and 16 scans per FID for a total duration of 1 h 46 min. Both the HNCO and HNCA were recorded at 298 K on an Oxford Instruments 600 MHz spectrometer equipped with a Varian Inova console and a 5 mm triple resonance room temperature probe with Hz-axis gradients.

The sensitivity-enhanced HNCO was recorded with 50 ($^{13}$C), 25 ($^{15}$N) and 1532 ($^{1}$H) complex points with 1256 Hz ($^{13}$C), 1056 Hz ($^{15}$N) and 8992 Hz ($^{1}$H) sweep widths using an interscan delay of 1 s and 8 scans per FID for a total duration of 13 h 10 min.

The sensitivity-enhanced HNCA was recorded with 40 ($^{13}$C), 30 ($^{15}$N) and 1532 ($^{1}$H) complex points with 2700 Hz ($^{13}$C), 1056 Hz ($^{15}$N) and 8993 Hz ($^{1}$H) using an interscan delay of 1 s and 16 scans per FID for a total duration of 24 h 55 min.

***Ubiquitin*** (76 residues, 8.5 kDa):
Sequence:
MQIFVKTLTGKTITLEVEPSDTIENVKAKIQDKEGIPPDQQRLIFAGK
QLEDGRTLSDYNIQKESTLHLVLRLRGG

All experiments were recorded at 298 K on a 500 MHz spectrometer equipped with a Varian console, as published previously[24]. The 2D $^{15}$N HSQC was recorded with 128 ($^{15}$N) and 1024 ($^{1}$H) complex points and 2000 Hz ($^{15}$N) 4000 Hz ($^{1}$H) sweep widths using an interscan delay of 1.3 s and 4 scans per FID for a total duration of 26 min.

The HNCO was recorded with 40 ($^{13}$C), 40 ($^{15}$N) and 1024 ($^{1}$H) complex points with 1500 Hz ($^{13}$C), 2000 Hz ($^{15}$N) and 3000 Hz ($^{1}$H) sweep widths using an interscan delay of 1.1 s and 8 scans per FID. The HNCA was recorded with 60 ($^{13}$C), 40 ($^{15}$N) and 512 ($^{1}$H) complex points with 4000 Hz ($^{13}$C), 2000 Hz ($^{15}$N) and 4000 Hz ($^{1}$H) sweep widths using an interscan delay of 1.3 s and 4 scans per FID.

***HSP16.5*** (113 residues, present as a dimer 24.8 kDa):
Sequence:
GSSSTGIQISGKGKGFMPISIIEGDQHIKVIAWLPGVNKEDIILNAVGDTLE
IRAKRSPLMITESERIIYSEIPEEEEIYRTIKLPATVKEENASAKFENGVLSVILP
KAESSIK

*Methanococcus jannaschii* heat shock protein 16.5 (HSP16.5) forms a predominant dimer (24.8 kDa) when truncated to the 'core' alpha-crystallin domain and shows many variable peak heights in spectra again consistent with subunit exchange. The HNCO was recorded at 323 K on an Oxford Instruments 600 MHz spectrometer equipped with a Varian Inova console and a 5 mm triple resonance room temperature probe with z-axis gradients. 50 ($^{13}$C), 25 ($^{15}$N) and 1298 ($^{1}$H) complex points were acquired with 1400 Hz ($^{13}$C), 1400 Hz ($^{15}$N) and 9000 Hz ($^{1}$H) sweep widths using an interscan delay of 1.5 s and 16 scans per FID for a total duration of 36 h 57 min.

***DDX4N*** (236 residues, 25.4 kDa):
Sequence:
MGDEDWEAEINPHMSSYVPIFEKDRYSGENGDNFNRTPASSSEMD
DGPSRRDHFMKSGFA
SGRNFGNRDAGECNKRDNTSTMGGFGVGKSFGNRGFSNSRFED
GDSSGFWRESSNDCEDN
PTRNRGFSKRGGYRDGNNSEASGPYRRGGRGSFRGCRGGFGLG
SPNNDLDPDECMQRTGG
LFGSRRPVLSGTGNGDTSQSRSGSGSERGGYKGLNEEVITGSGKN
SWKSEAEGGES

1–236 residues of human DDX4 protein (sequence termed DDX4N1[39]).

DDX4N experiments were performed at 303 K on an Oxford Instruments 750 MHz spectrometer equipped with a Bruker Avance III HD console and a 5 mm TCI CryoProbe with z-axis gradients. The 2D $^{15}$N-$^{1}$H BEST-TROSY HSQC was recorded with 128 ($^{15}$N) and 1024 ($^{1}$H) complex points and respective sweep widths of 1597 Hz and 9803 Hz using an interscan delay of 0.2 s and 32 scans per FID for a total duration of 55 min. The 3D BEST-TROSY HNCO was recorded with 64 ($^{13}$C), 64 ($^{15}$N) and 1024 ($^{1}$H) complex points and 2832 Hz ($^{13}$C), 1597 Hz ($^{15}$N) and 9804 Hz ($^{1}$H) sweep widths using an interscan delay of 0.2 s and 32 scans per FID for a total duration of 3 h 25 min.

The 3D BEST-TROSY HNCA was recorded with 64 ($^{13}$C), 64 ($^{15}$N) and 1024 ($^{1}$H) complex points and 5291 Hz ($^{13}$C), 1597 Hz ($^{15}$N) and 9804 Hz ($^{1}$H) sweep widths using an interscan delay of 0.2 s and 156 scans per FID for a total duration of 16 h 6 min. All spectra were processed using the UnidecNMR software, which relies on NMRPipe and nmrGlue.

## Generalised instructions to run UnidecNMR

For best performance, we recommend a Lorentz-to-gauss window function in all dimensions, which will result in a peak shape that well matches a pseudo-voigt function (Eq. 2). Empirically, care to minimise long Lorentzian tails tends to generate optimal results. This choice is not essential, other window functions, including exponential and sine-bell can be used. In practice, our experimental dataset was processed using a small range of apodization functions (Supplementary Table 2) and results are largely independent of this choice.

The next step is to obtain general peak shape parameters for use as the peak shape filter. There are a range of ways to do this. First, a peak shape can be 'guessed', a trial UnidecNMR calculation is performed, the result inspected, then the values adjusted based on whether the result has obvious over or under picked the spectrum. Second, the UnidecNMR GUI loads in the most intense peaks detected, allowing a user to either manually or algorithmically fit them within the software (the 'Fit Peaks' tab). Sliders allow the various parameters to be adjusted until the shape is as desired.

Once a peak shape has been determined, UnidecNMR can be run, the result inspected, then the peak shape further tweaked until the simulated spectrum well resembles the original and the positions of the selected peaks look reasonable. The overlay of the reconvolved spectrum and the original, as presented in the GUI, make this a straightforward exercise (Figs. 1, 3, 5–7, S2, S7). Either the projections in 3D/4D data or the overlay of the 3D/4D data specifically can be compared in this way.

Finally, a noise threshold has to be set. A useful protocol for this is to inspect the data in the projections window, place the lower contour in a desirable position, and then press 'set'. In practical applications, we typically run with a high threshold to enable a rapid computation allowing assessment of the peak shape, before honing by lowering the threshold to a level suitable to detect all relevant spectral features.

The number of CPUs can be set to any value, and the calculation will be parallelised at the level of the Fourier transform via FFTW3. Improvements in calculation time are achieved, but owing to the complexities of parallelising Fourier transforms, reduction in calculation time will not be linear with the number of CPUs.

When running UnidecNMR, there are two further considerations. The first is the convergence of the algorithm. Two numbers can be selected: the maximum number of iterations and the convergence threshold. These can be set in the GUI to 'quick', 'medium' and 'accurate'. Increasing the maximum number of iterations and decreasing the convergence threshold will result in a longer but more thorough calculation. All results shown in the paper were achieved with either 'medium' or 'accurate' settings, but when initialising a calculation, 'quick' settings are helpful.

The values for convergence of the algorithm (maxIter, convergence) were set as 'quick' $(25, 10^{-5})$, 'medium' $(50, 10^{-7})$ and 'accurate' $(100, 10^{-8})$. In calculations shown in this paper, the maximum number of iterations typically halts the calculation. The final peak list tends to vary slightly between 'quick' and 'medium', and rarely varies when comparing 'medium' and 'accurate'. In cases where the algorithm performance seems poor, the maximum number of iterations should be increased.

Finally, as described in the text, the optimiser has one further user scalable parameter, 'fac'. We find excellent results for 2D data with fac = 1.4, and for 3D, 1.6 when using the 'boring' mode. If the program is missing peaks that are highly overlapped, decrease fac. If it picks too many, increase it. The settings become highly intuitive after running the program a small number of times.

### Reporting summary

Further information on research design is available in the Nature Portfolio Reporting Summary linked to this article.

## Data availability

Benchmarking data is available for download from http://UnidecNMR.chem.ox.ac.uk. The 2D HSQC and 3D HNCA/HNCO spectra from the four proteins in the benchmark (ubiquitin, aB-crystallin, hsp27 and DDx4) are present, with input files that allow their processing and peak picking to be performed precisely as described in this manuscript (in conjunction with the program, see "Code availability" statement). These act as a template to allow users to adapt their own data to our environment and as a validation that the software works as described in this manuscript. Experimental details for the different systems and spectrometer acquisition settings are found in the "Methods" section. Supplementary Table 2 describes detailed settings for processing the FIDs into spectra. Supplementary Table 1 describes how the different peak-picking methods were scored.

## Code availability

Software and benchmarking data are available for download from http://UnidecNMR.chem.ox.ac.uk. The core algorithm is written in C++ and will be available in pre-compiled binary form. The Python code will be distributed. This provides a GUI allowing access to the processing functions of nmrPipe, a 1/2/3/4D spectrum viewer, from which the peak-picking functions of the UnidecNMR can also be directly accessed. The Software is distributed "AS IS" under this Licence solely for non-commercial use. If you are interested in using the Software commercially, please contact the technology transfer company of the University to negotiate a licence. Contact details are:

"enquiries@innovation.ox.ac.uk". We will thoroughly welcome community input both in improving the user experience and expanding the benchmark to enable future development and improvement of computational tools such as these.

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

## Acknowledgements

Many thanks to the following for providing the data for this project: Reid Alderson provided the HSP27 data; The Clore laboratory provided the 4D dataset of methyl labelled EIN; The Kay laboratory provided the 3D dataset of methyl labelled ATCase. Many thanks to Richard Harris and Paul Driscoll for the Ubiquitin NMR resource. Thank you to Aziz Khan for preparing the 2,3-sialyllactose sample. Reid Alderson also kindly provided comments on the manuscript. Thanks also to Pembroke College, Oxford, for supporting this project. A.J.B. has received funding from the European Research Council (ERC) under the European Union's Horizon 2020 research and innovation programme (grant agreement No 101002859).

## Author contributions

Materials and raw data: O.T., M.B., G.K., T.J.N., C.R. Algorithm initial design: M.T.M., A.J.B. Algorithm implementation, development, benchmarking and testing: C.B., A.J.B. The manuscript was written by C.B. and A.J.B. with input from all authors.

## Competing interests

The authors declare no competing interests.
