## [Transparent Peer Review file · Nature Communications]

UnidecNMR: Automatic peak detection for NMR spectra in 1-4 dimensions

Corresponding Author: Professor Andrew Baldwin

Parts of this Peer Review File have been redacted as indicated to remove confidential information.

Version 0:

Reviewer comments:

Reviewer #1

(Remarks to the Author)

Overall, the paper describes a new and, from the data provided, highly effective method of peak picking multidimensional nmr spectra in an automated manner. The results and method appear to be sound and generally the data proving the point is good. However, the explanation of how the algorithm works is a little sparse in places on the details... The need for good nmr peak pickers for use in the software used by the community remains an important topic as poor peak pickers cause much time to be lost to manual peak analysis in the community. The one area that the paper doesn't cover well and which I would urge the authors to address is providing information on the quality of the peak picking in terms of terms of position and intensity as their main metric is quantity of peaks picked.

There are (quite a number) of minor matters that could do with addressing which are detailed below

1. Page 2 Figure 1 is somewhat cramped and the labels are tiny
2. Page 4 para 1 a spectrum with sugars is discussed but no figure is referenced
3. Page 4 para 2 The range of molecular weights discussed doesn't really include spectra with challenging properties due to rmm it would have been helpful if the example triple resonance spectra included spectra from proteins in the 20-30kDa range
4. Page 4 para 3 the list of peak picking algorithms listed is not consistent with figure 2 which includes dumpling which isn't listed here. WaVPeak is spelt inconsistently compared with later in the text
5. Page 4 para 3 The referenced figure 6 doesn't show that the noesy spectra are superior it just mentions it in the figure text. Also, the authors should define superior, the complete text really only deals with improving peak numbers and doesn't deal with quality (reliability of position and intensity), it should be made clear that this is the case early in the text and some information on recovered intensities and positions would improve the text.
6. Page 4 para 3 here is the improvement in results with symmetry and 2D spectra shown if its later in the text that should be indicated or if there is a relevant figure it should be quoted.
7. Page 4 para 3 'is completed in under a minute for a 3D' ...but on what sort of computer, from the distribution an M1 mac ? It may be better to be more specific and talk about processors, memory and cpu speeds. Also is the program run on a single thread or multiple threads in these comparisons? Modern machines can have many more threads available... Indeed, the distributed software includes `decon_parallel_Darwin_arm` which points towards a parallel implementation but what methodology was used and how well does it scale with thread count?
8. Page 4 para 3 The package has been tested on Windows, Mac, and Linux, however the software is only distributed as a mac version which is disappointing...
9. Page 5 Figure 2 leaves several question unanswered. The authors discuss 7200 spectra but there is no detailed discussion how they were distributed between s/n and peak separation, what the ranges used were (though this can be guessed off the axes and resolution in the contour plots) and what the separations were. Were there repeats with different random number seeds (see more below) so some statistics and repeatability of results could be judged (and could they show this data). What's are the units of separation? What peak shape was used. The relation of the positions of the crosses on the figures and the numbers used on the figures should be pointed out. Is FWHM defined somewhere (presumably its full width half maximum). The set of program results shown is not consistent with the text and not all programs listed in the main

text are used. How repeatable are the results with respect to repeats with different random number seeds... it is possible to be selective with results with respect to random noise (not that I am suggesting the authors are!) and this technique has been used to achieve better astronomical images in practical approaches (I think this is possibly answered by the scale on panel c which suggests 100 repeats which would give a grid of 8x9) What is the blue line present on some of the contour plots but not others representing? Why is dumping on the right hand panel of 3 in panel counted as a failure? How do the panels in b relate to panel a. Why are some peak pick crosses in different colours to others? The upper right panel of figure 2 says NMRNet but this isn't one of the peak pickers used in the rest of the figure.

10. Page 6 It is not clear why the results start with theory

11. Page 6 Figure 3 the ii on the expansion panel isn't repeated on b's expansion panel either describe the labels in the figure legends or delete them and mention the expansions

The meaning of variations in peak shape with respect to peak 6c are not clear to this reader. What's the information on the samples, e.g what concentrations were the samples, what was the spectrometer frequency, was this in D₂O/H₂O 95%: 5% (one might presume so for the first spectrum where excitation sculpting was used), what was the resolution, what processing was applied including window functions etc, how many scans were measured. This is all important information if a user wants to reproduce the result.

12. Page 7 para 1 The theory section says nothing about how the order of computation on peaks (largest first in the same way as peaks are picked in the CLEAN / SCRUB NUS algorithm?) Does the order of peaks used in the calculation affect the result. Are calculations made on groups of peaks which are overlapped or all peaks at once

13. Page 8 The implementation of the algorithm isn't entirely clear

a. What algorithm does the squash clustering phase use and is the result dependant on the algorithm.

b. Does the algorithm just carry out the 2 steps just make a single pass or there is some iteration, if so how is this implemented?

c. Is the data in 3d data clustered or is it applied a whole nD dataset

d. How is s/n defined, its mentioned here but only defined later and there is more than one method...

e. How is convergence defined

14. Page 8 para 2 The optimal figures for fa and squash in the text don't appear to match those in figure S4.

15. Page 8 para 4 Fig 3d doesn't exist

16. Page 8 para 3 It remains reasonable outside these limits... once again this is presumably referring to the number of peaks measured rather than their fidelity, the authors should be clear on what counts as success.

17. Page 9 para 1 When referring to NMRNet what do the authors mean by 'real' data. Also the figure labels here are confusing as mention is made of figure 3 with is not a comparison between methods but a set of 1d spectra picked with UnidecNMR

18. Page 9 para 2 'The algorithm can be run iteratively and compared to the raw data' it isn't entirely clear what the authors mean at first reading. On further reading what they appear to mean is that several manual runs can be made of the UnidecNMR algorithm with different peak shape parameters. The use of iteratively could imply that this can be measured automatically...

19. Page 9 Figure 4

a. 'boring' mode is often referred to as templated or restricted peak picking (cf the sparky manual page at <https://www.cgl.ucsf.edu/home/sparky/manual/manual.html>) it would be better to use terms more commonly used by the nmr community

b. Is the set of 4 over all the spectra or just the hsqcs?

20. Page 10 para 2 Paper 25 doesn't really show any nmr spectra but the text makes comments about line widths etc...

21. Page 12 para 1 Throughout the script Å is used for angstrom rather than the correct symbol Å

22. Page 12 para 2 it would be good indicate the signals believed to be from overly intense peaks due to dynamics on figure 6a

23. Figure 6 states that 60% more peaks were identified for ATCase and Ein, is this across the pair or were they both 60? It would be better to state the values individually

24. Page 14 para 3 If possible the downloadable test data should not be on the groups or home institutions website but somewhere with a more long term storage guarantee such as github or the BMRB

25. Page 15 para 2 The authors use and cite the ubiquitin resource but this resource appears to no longer be accessible online

26. Page 15 para 2 The authors should note they are missing an ERC grant numbers in the Acknowledgements

27. Page 20 Table 1 can the authors define 'low intensity' in para 5

28. Page 21 para it would help in reading the table if the definition of A/B/C/D appeared earlier in the caption

29. Page 21 para 4 Sparky which can only perform 2D Peak Picking

30. Page 22 Fig S1 the reference to figure 3 is erroneous again and some isotope numbers are not super scripted here and in other places within the text

31. Page 23 Fig S3 the pictures of the gui are much too small to be useable a figure per page would be more reasonable if there is room in supplementary

32. Page 23 Fig S2 the peak label crosses hide the quality of the reconstruction

33. Page 24 Figure S4 y axes are labelled s:n ratio but without a value. It should be made clear that the numbers on the upper figures refer to the left contour plots (in this and other places). The best results are indicated by a cross that defines the best result and its position appears to be inconsistent with earlier in the text.

The software was installed and tested and worked as described and the demo videos were accurate and useable.

The authors should note that the way they recommend installing the software is not current best practice and may cause

some users problems with their python installations, and fail on older mac computers. The referee would suggest they look at pypi pipx and pyscaffold as routes for distribution.

Reviewer #2

(Remarks to the Author)

Buchanan et al. present a new computer program called UnidecNMR for NMR peak picking. Their testing was performed on 1D to 4D NMR spectra. Authors claim UnidecNMR outperforms a number of freely available algorithms and produces similar quality of manually picked peaks.

Peak picking is a certainly one of the most challenging tasks in biomolecular NMR.

However, it is unclear to me if UnidecNMR is a novel work and really a game changer because of a few things.

- As authors already wrote in the manuscript, there are already a few peak picking programs.
- Using 2D peak list restraints sound like the restricted peak picking method which is already a common method in the field. Authors should clarify why it is different and novel.
- What is reflection symmetry? Description is slim. It sounds like just seeing presence of a signal in the symmetric position.
- Methods represented in here is very similar to ChiFit and Newton introduced decades ago (Chylla et al.). They read NMRPipe processing parameters for time-domain data simulation and fitting in addition to peak shape fitting on the frequency-domain data. FuDa and PINT are well-known peak shape fitting programs, and they have not been discussed and benchmarked while the UnidecNMR does peak shape fitting for deconvolution.

Supporting only nmrPipe's file is also a limiting factor. Currently, many scientists cannot use nmrPipe due to technical difficulties and have moved to TopSpin, MNova, etc for data processing. Therefore, nmrPipe is not always in their pipeline any more. When the peak picking is performed, it is very likely that users in the stage of using analysis programs like Sparky/Poky/CARA/CCPN/NMRView/ETC. Authors should be aware of that.

Benchmark must illustrate more challenging problems. For example, data on figure 2 will be analyzed by a simple local maximum method easier, faster and more accurate not to mention analysis programs all have that algorithm alongside user-friendly interactive user interfaces for performing and improving by putting users' insights. Such as 3D C13-NOESY and 2D-CC from a protein will be good candidates.

Reviewer #3

(Remarks to the Author)

The manuscript describes a peak picker which relies on deconvolution of peaks. This is a very established approach and does not have any novelty. Yet, the authors implement it in a fashion that outperforms other algorithms and also even experienced spectroscopists according to the presented results.

I would like to state that it is impossible for a reviewer to judge these claims and in other fields the performance of algorithms are judged in competitions for example regarding structure prediction and so on. An algorithm which is not published and available cannot be subjected to such a test by the community. The results look plausible to me and therefore the manuscript and the algorithm should be published.

Still there is need for clarification and better description in parts. In Fig. 2C there is a comparison of the various algorithms regarding signal to noise and peak separation. It appears that peaks 1,4,7 are at the limit for reliably being picked by PICKY while just still reliably picked by UNIDEC. How do peaks look like that UNIDEC does not pick correctly? They should be included in Fig. 2 to give a feeling to the experienced spectroscopist where UNIDEC has its limits.

The peak shape g that the user needs to provide is conveniently derived how?

There are two empirical factors fac and $squash$ which modify the peak shape and optimized values are given for them: ($fac=1.6$, $squash=0.625$) Are these values universal? Or do they need to be adjusted for each spectrum? If (and this is my impression from the manuscript) they are universal, the authors might provide some insight why these values are optimal and how that could be explained given the mathematical background of the procedure. The authors write: "Below these thresholds the success of the algorithm falls away from 100%, and its success depends on the exact shape of the noise profile in an individual spectrum. It nevertheless remains reasonably successful outside these windows, and typically fails in cases where an experienced user would also be unconfident." It would be great if the authors could provide for example in the SI such examples that the experienced spectroscopist can convince himself of the signal quality under these circumstances.

In Fig. 4 there are comparisons on 2D spectra and UNIDEC outperforms all comparisons. Specifically the number of wrongly identified peaks is much smaller in UNIDEC than in the other programs. Still UNIDEC did wrong peaks and it would be interesting to see what the features of these peaks are. Would they be identified by an assignment program working with peak lists such as MARS or FLYA because some spurious noise was picked but the peaks would not lead to wrong assignments?

Regarding the discussion of Fig. 3, is it right to assume that the 1D peak picking of multiplets does identify i.e. a doublet of doublets as such but just identifies 4 lines? Are the peak positions so accurate that one could assemble e.g. a doublet of doublets from the fact that frequency difference of peak 1 and 2 should be the same as 3 and 4; and that frequency difference of peak 1 and 3 should be the same as 2 and 4, maybe taking the intensity (or integral which is available since the peak

shape is defined) into account additionally? Along the same lines, would it be obvious from integrals which peaks belong to the same molecule in case of mixtures with different concentrations?

Regarding the statement: "One spectrum of note was an HNCO acquired on Hsp16.5(unpublished data), where the decoupler was mis-set, resulting in a triplet for each peak in the ^{13}C dimension (Fig. S1b). By increasing the effective peak width used for calculations, we were able to account for this deficiency. By contrast, the other algorithms found analysing this specific experiment highly challenging." The authors should answer the following questions: In the 1D case, multiplets were assigned. Why could such an approach not be used to here also given the fact that the couplings are known: 7 and 11 Hz is the splitting which could be incorporated in the peak shape g. The remark "this specific experiment highly challenging" needs a more scientific explanation.

In the case of the methyl NOESY, 60% more peaks were found than by an experienced user. It would be great to see some of those additional peaks in the spectrum which were missed by an experienced user. Fig. 6 does not really report this information. Is it because of signal to noise or because of overlap of resonances? Some peaks NOESY peaks refer to distances a little larger than 14 Angstrom. For loops whose dynamics was not included in the pdb file of reference this makes sense. Do all those peaks with long distances belong to the loop? 10 Angstrom is quite a long distance. What mixing time was used in that NOESY? In addition, in Fig. 6iii there are quite a number of peaks with relative cross peak intensity of 0. How can these peaks be found if their relative and then necessarily absolute intensity is 0/close to 0?

The ERC grant number is not stated in the acknowledgment.

Version 1:

Reviewer comments:

Reviewer #1

(Remarks to the Author)

I think the authors have done a very thorough job of answering this reviewers comments and am happy with paper in its latest form and would recommend publication

Reviewer #2

(Remarks to the Author)

1. The provided link for UnidecNMR in the manuscript does not work.
2. "Nature Comm." is open access. Is the UnidecNMR itself open source? It appears that only the wrapper to call the program is open source from the given code. The authors have provided a compiled version, so I cannot evaluate it. Additionally, the HNCO and HNCA examples, claiming superiority, haven't been provided. This is necessary because the "peak-picking" process can be biased by different parameters like contour settings.
3. Is this a decomposition or deconvolution? This work seems to be neither of those but rather filtering. Please elaborate.
4. Taking raw FIDs directly can be a drawback. NMRPipe, TopSpin, and NMRFX Processor programs have sufficient functionalities to improve the quality of the data, and users will not be able to benefit from those programs in their pipelines.
5. Again, the authors insist on UnidecNMR's great performance. Please provide specific instructions or examples for how I can test UniDecNMR's performance.

Reviewer #3

(Remarks to the Author)

all my comments are taken care of adequately.

Version 2:

Reviewer comments:

Reviewer #2

(Remarks to the Author)

The reviewer recommends thorough real-life testing by authors before submission. The UnidecNMR program currently suffers from performance issues and bugs, rendering it unusable. However, the benchmark results suggest some potential for the underlying algorithm. To improve usability, the reviewer suggests removing the GUI entirely and focusing on a simple command-line interface (CLI) tool.

Following are our point-by-point responses to the 3 reviewers. Comments from us are in yellow, specific entries into the text, where appropriate, are in blue. These accompany the revised manuscript where we have indicated in yellow the areas where we have made substantial changes.

Reviewer #1 (Remarks to the Author):

Overall, the paper describes a new and, from the data provided, highly effective method of peak picking multidimensional nmr spectra in an automated manner. The results and method appear to be sound and generally the data proving the point is good.

We thank the reviewer for their comments and this particularly detailed review. The manuscript is much improved by our working through this list of corrections.

However, the explanation of how the algorithm works is a little sparse in places on the details...

We do suspect here, based on later questions, that the reviewer is assuming the algorithm is more complex than it is: the algorithm at its core really is very simple and is operating exactly as we describe it in the theory section. Should anyone write these equations into python, or their favourite language, as written, they will get same performance as described. There are no random numbers, the output is completely deterministic. It's really very neat!

There is not need to, for example, rank the peak heights as later assumed, and have the algorithm make additional decisions of the type they assume that we must be doing. We have embellished our description of the algorithm and its implementation in the revised manuscript.

The need for good nmr peak pickers for use in the software used by the community remains an important topic as poor peak pickers cause much time to be lost to manual peak analysis in the community. The one area that the paper doesn't cover well and which I would urge the authors to address is providing information on the quality of the peak picking in terms of terms of position and intensity as their main metric is quantity of peaks picked.

To further assess this, we have added supplementary figures 8 and 9. In fig S8, we compare the performance of the various peak pickers on simulated 1D data. In short, the accuracy of the peak position depends on the signal to noise ratios, becoming more accurate the higher the S/N. Provided resonances were separated by 1.25 times the FWHM, we see excellent reproduction of the intensities (Fig. S9).

For the 'real' experimental data we cannot do a similar test, because 'true' positions and intensities are essentially unknown. The best we can do is infer from the manually picked NMR spectra.

There are (quite a number) of minor matters that could do with addressing which are detailed below

We thank the reviewer for taking the time to do a detailed dive and to annotate these, the fixing of which certainly improves the manuscript.

1. Page 2 Figure 1 is somewhat cramped and the labels are tiny

We have adjusted this.

2. Page 4 para 1 a spectrum with sugars is discussed but no figure is referenced

Fixed.

3. Page 4 para 2 The range of molecular weights discussed doesn't really include spectra with challenging properties due to rmm it would have been helpful if the example triple resonance spectra included spectra from proteins in the 20-30kDa range

AlphaB crystallin and HSP27 both form ca. 20kDa dimer, and so most of our spectra fall in this range. We would also say that being large doesn't make a protein challenging: exchange broadening is the bigger challenge. And a significant proportion of the resonances from AB-crystallin are exchange broadened rendering this challenging.

We have now also included the 24kDa intrinsically disordered protein as a challenging test to further address this concern (See letter).

4. Page 4 para 3 the list of peak picking algorithms listed is not consistent with figure 2 which includes dumping which isn't listed here. WaVPeak is spelt inconsistently compared with later in the text

We have fixed this.

5. Page 4 para 3 The referenced figure 6 doesn't show that the noesy spectra are superior it just mentions it in the figure text. Also, the authors should define superior, the complete text really only deals with improving peak numbers and doesn't deal with quality (reliability of position and intensity), it should be made clear that this is the case early in the text and some information on recovered intensities and positions would improve the text.

We have clarified this in the text. Briefly, we see 294/420(714 total) NOES from ATCase/EIN from previous work as cited, and using our algorithm we pick 460/660 (1120 total), a 60% increase as stated. From the intensity/distance curves, the picked NOEs are demonstrated to be consistent with the known structures.

We previously said this in the text:

"The intensity of the resonances was plotted against the known distance spanned in the protein to determine if the picked NOEs are consistent with the expected structural forms (1d09³¹/1eza³² ATCase/Ein) (Fig. 6a). Both plots revealed the vast majority of the NOEs to be within 10A and so are consistent with the known structures (Fig. 6a) and have a similar pattern to those picked manually ²⁹."

We have clarified to note:

"460 pairs of cross peaks were identified for ATCase and 660 for Ein, which combined is 60% more than obtained from a manual analysis (294/420 respectively, used in previous work²⁹)."

6. Page 4 para 3 here is the improvement in results with symmetry and 2D spectra shown if its later in the text that should be indicated or if there is a relevant figure it should be quoted.

A link to the NOE figure has been added.

7. Page 4 para 3 'is completed in under a minute for a 3D' ...but on what sort of computer, from the distribution an M1 mac ? It may be better to be more specific and talk about processors, memory and cpu speeds. Also is the program run on a single thread or multiple threads in these comparisons? Modern machines can have many more threads available... Indeed, the distributed software includes decon_parallel_Darwin_arm which points towards a parallel implementation but what methodology was used and how well does it scale with thread count?

We have clarified to:

The time of the calculation depends on the size and dimensionality of the data, but it completes in 5-30s for a 2D spectrum, 30-120s for a 3D, and several minutes for a 4D on a 2021 MacBook Pro equipped with an M1 Pro processor and 16GB of RAM.

On parallelisation, this is accomplished using FFTW3 at the level of the Fourier transform. This does not give a simple or linear speed enhancement, but empirically calculations are faster when giving more cores owing to the complexity of the algorithm. We have briefly explained this now in the text:

The number of CPUs can be set to any value, and the calculation will be parallelised at the level of the Fourier transform via FFTW3. Improvements in calculation time are achieved, but owing to the complexities of parallelising Fourier transforms, reduction in calculation time will not be linear with the number of CPUs.

8. Page 4 para 3 The package has been tested on Windows, Mac, and Linux, however the software is only distributed as a mac version which is disappointing...

As noted earlier, we have now generated windows, mac intel, mac ARM and Linux distributions.

9. Page 5 Figure 2 leaves several question unanswered. The authors discuss 7200 spectra but there is no detailed discussion how they were distributed between s/n and peak separation, what the ranges used were (though this can be guessed off the axes and resolution in the contour plots) and what the separations were. Were there repeats with different random number seeds (see more below) so some statistics and repeatability of results could be judged (and could they show this data). What's are the units of separation? What peak shape was used. The relation of the positions of the crosses on the figures and the numbers used on the figures should be pointed out. Is FWHM defined somewhere (presumably its full width half maximum). The set of program results shown is not consistent with the text and not all programs listed in the main text are used. How repeatable are the results with respect to repeats with different random number seeds... it is possible to be selective with results with respect to random noise (not that I am suggesting the authors are!) and this technique has been used to achieve better astronomical images in practical approaches (I think this is possibly answered by the scale on panel c which suggests 100 repeats which would give a grid of 8x9)

We have clarified precisely how we conducted the numerical simulations in both the legend of Fig. 2 and in the extended methods. Specifically on the matter of the random number generator, we repeated the noise simulation 100 times for each FWHM/SN combination, and the results shown reflect the results generated from the statistical average over all of these.

What is the blue line present on some of the contour plots but not others representing?

This is the contour line indicating where the relevant algorithm was 100% correct. We have clarified in Fig legend 2.

Why is dumpling on the right hand panel of 3 in panel counted as a failure?

NMRNet has picked two peaks here not just one. We have made this clear now by marking the two peaks in different colours. This was not clear before and thank you for pointing this out!

How do the panels in b relate to panel a. Why are some peak pick crosses in different colours to others? The upper right panel of figure 2 says NMRNet but this isn't one of the peak pickers used in the rest of the figure.

We have clarified: also, Dumpling and NMRNet are in fact the same. Dumpling is the name of the GUI written around NMRNet, whereas NMRNet is the peak picker.

10. Page 6 It is not clear why the results start with theory

The reviewer noted that they would like a description of the algorithm: but this is precisely where we describe its action, in the theory section.

11. Page 6 Figure 3 the ii on the expansion panel isn't repeated on b's expansion panel either describe the labels in the figure legends or delete them and mention the expansions
The meaning of variations in peak shape with respect to peak 6c are not clear to this reader. What's the information on the samples, e.g what concentrations were the samples, what was the spectrometer frequency, was this in D2O/H2O 95%: 5% (one might presume so for the first spectrum where excitation sculpting was used), what was the resolution, what processing was applied including

window functions etc, how many scans were measured. This is all important information if a user wants to reproduce the result.

We have now included a detailed table in the supplementary information. Full descriptions will be available for download for future academic use.

12. Page 7 para 1 The theory section says nothing about how the order of computation on peaks (largest first in the same way as peaks are picked in the CLEAN / SCRUB NUS algorithm?) Does the order of peaks used in the calculation affect the result. Are calculations made on groups of peaks which are overlapped or all peaks at once

There is no ordering in the algorithm of this type, it proceeds exactly as described: the algorithm starts from the initial condition that intensities match the spectrum intensities. The core algorithm iteratively adjusts all intensities according to equation (1), exactly as described, until it converges. This is the Unidec algorithm as published previously in essence. The clustering steps added for UnidecNMR are described in detail in the text.

13. Page 8 The implementation of the algorithm isn't entirely clear

We have described every step and put in the relevant equations. What we have here really is sufficient for someone who likes coding that wishes to implement our algorithm. As far as we can judge from the questions here the reviewer suspects that there are more steps that we are not showing. There really are not. We would suggest people use our implementation in C++ and FFTW but really people are free to write their own.

a. What algorithm does the squash clustering phase use and is the result dependant on the algorithm.

We are using the simplest possible clustering algorithm. Where two peaks are close to within a threshold, which we empirically determine to be ca. 0.75 of the FWHM of the peak function being used to analyse the data (See text and supplementary information), all intensity is transferred from the least intense positions, to the most intense ones. This is described in the text, and the effects are shown in detail in the supplementary information.

b. Does the algorithm just carry out the 2 steps just make a single pass or there is some iteration, if so how is this implemented?

The implementation follows equation (1), exactly as stated. The changes made in each pass are recorded, and this is used as a convergence metric. We have added a description of convergence to the text

"To determine convergence, the changes made in intensities are assessed in each step, and when these fall below a user specified threshold, or when the calculation exceeds a pre-specified number of maximum iterations, the calculation stops."

In the supplementary usage notes we now say this on convergence:

Two numbers can be selected, the maximum number of iterations, and the convergence threshold. These can be set in the GUI to 'quick', 'medium' and 'accurate'. Increasing the maximum number of iterations and decreasing the convergence threshold will result in a longer but more thorough calculation. All results shown in the paper were achieved with either 'medium' or 'accurate' settings, but when initialising a calculation, 'quick' settings are helpful.

The values for convergence of the algorithm (maxIter, convergence) were set as 'quick' (25, 10^{-5}), 'medium' (50, 10^{-7}) and 'accurate' (100, 10^{-8}). In calculations shown in this paper, the maximum number of iterations typically halts the calculation. The final peak list tends to vary slightly between 'quick' and

'medium', and rarely varies when comparing 'medium' and 'accurate'. In cases where the algorithm performance seems poor, the maximum number of iterations should be increased.

c. Is the data in 3d data clustered or is it applied a whole nD dataset

We apply exactly as stated: the entire dataset is analysed in one go. Which means lots and lots of fourier transforms, which using FFTW, are fast even for 4D data. The data could be stripped down into cubic sections to speed the algorithm up. But it's already fast we think, so we haven't done this. A user is free to extract restricted regions of their spectra if they want to speed things up.

d. How is s/n defined, its mentioned here but only defined later and there is more than one method...

We are taking maximum signal divided by standard deviation of the noise. We have stated this more clearly when describing our synthetic tests of the algorithm in the manuscript.

e. How is convergence defined

See above. We have expanded our description of this in the manuscript. In brief, the total summed intensity in each iteration is monitored. When the difference in intensity between iterations is below a threshold, the program stops. A user can also specify a maximum number of iterations..

14. Page 8 para 2 The optimal figures for fa and squash in the text don't appear to match those in figure S4.

We have fixed this.

15. Page 8 para 4 Fig 3d doesn't exist

We have fixed this.

16. Page 8 para 3 It remains reasonable outside these limits... once again this is presumably referring to the number of peaks measured rather than their fidelity, the authors should be clear on what counts as success.

We have fixed this.

17. Page 9 para 1 When referring to NMRNet what do the authors mean by 'real' data. Also the figure labels here are confusing as mention is made of figure 3 with is not a comparison between methods but a set of 1d spectra picked with UnidecNMR

We have clarified this in the text. 'Real' data comes from an actual experiment. The alternative is simulated data, which we made as described. We have clarified this distinction.

18. Page 9 para 2 'The algorithm can be run iteratively and compared to the raw data' it isn't entirely clear what the authors mean at first reading. On further reading what they appear to mean is that several manual runs can be made of the UnidecNMR algorithm with different peak shape parameters. The use of iteratively could imply that this can be measured automatically...

We agree in principle. In the software, the code can make a good 'first guess' of the peak shape using an automated routine. Best results however come from looking at these results and making manual adjustments as necessary. For example, if too many little peaks get picked, raise the threshold. If it looks like peaks are being picked in the edges of the peak, then increase the width of the filter. In practise, this isn't arduous. We have expanded our description of this in the text. In short, we would say our automatic routines for peak shape work pretty well, but our method still benefits from having a user look at the result to double check things are sensible.

19. Page 9 Figure 4

a. 'boring' mode is often referred to as templated or restricted peak picking (cf the sparky manual

page at <https://www.cgl.ucsf.edu/home/sparky/manual/manual.html>) it would be better to use terms more commonly used by the nmr community

We talked to 5 different biological NMR groups in the UK and none knew this term. There are some significant differences between how UnidecNMR works and how sparky's restricted peak picking work. We like the term boring, find it physically intuitive, and so we prefer to use this. We discuss this in the text.

b. Is the set of 4 over all the spectra or just the hsqcs?

We have fixed this. This is over all spectra. Individual breakdowns are included in the supplementary information.

20. Page 10 para 2 Paper 25 doesn't really show any nmr spectra but the text makes comments about line widths etc...

21. Page 12 para 1 Throughout the script A is used for angstrom rather than the correct symbol Å

Now fixed.

22. Page 12 para 2 it would be good indicate the signals believed to be from overly intense peaks due to dynamics on figure 6a

We have indicated this.

23. Figure 6 states that 60% more peaks were identified for ATCase and Ein, is this across the pair or were they both 60? It would be better to state the values individually

See earlier: we have clarified: this is 60% more compared for both, and we state the specific values.

24. Page 14 para 3 If possible the downloadable test data should not be on the groups or home institutions website but somewhere with a more long term storage guarantee such as github or the BMRB

We agree. We will put this on github. The BMRB does not storage data in a manner that fits what we need. We are in discuss with Jeff Hoch and the BMRB team about finding a better way to do this so we can have databases for the purpose of training peak pickers.

25. Page 15 para 2 The authors use and cite the ubiquitin resource but this resource appears to no longer be accessible online

There is a reference, and it was available when we downloaded it! We will distribute the ubiquitin data with our benchmark with appropriate citations.

26. Page 15 para 2 The authors should note they are missing an ERC grant numbers in the Acknowledgements

Fixed!

27. Page 20 Table 1 can the authors define 'low intensity' in para 5

Now fixed.

28. Page 21 para it would help in reading the table if the definition of A/B/C/D appeared earlier in the caption

Now fixed.

29. Page 21 para 4 Sparky which can only perform 2D Peak Picking

Now fixed.

30. Page 22 Fig S1 the reference to figure 3 is erroneous again and some isotope numbers are not super scripted here and in other places within the text

We have edited these to make them consistent.

31. Page 23 Fig S3 the pictures of the gui are much too small to be useable a figure per page would be more reasonable if there is room in supplementary

The resolution of the figures is high, so we respectfully suggest the use of the zoom function present in most browsers. The data is supplied with UnidecNMR so users can reproduce this themselves on their own computers should they wish. The youtube videos we feel are a better tutorial than we can achieve in the text.

32. Page 23 Fig S2 the peak label crosses hide the quality of the reconstruction

Now fixed. The reconstructions to our eyes are extremely close to the raw data, making it generally hard to distinguish. We would like to point out that this helps make the point that the algorithm is doing a remarkably good job.

33. Page 24 Figure S4 y axes are labelled s:n ratio but without a value. It should be made clear that the numbers on the upper figures refer to the left contour plots (in this and other places). The best results are indicated by a cross that defines the best result and its position appears to be inconsistent with earlier in the text.

Now fixed.

The software was installed and tested and worked as described and the demo videos were accurate and useable.

We are delighted to hear this.

The authors should note that the way they recommend installing the software is not current best practice and may cause some users problems with their python installations, and fail on older mac computers. The referee would suggest they look at pypi pipx and pyscaffold as routes for distribution.

We have looked at this and we thank the referee for their thought and contributions to our paper. The paper is enhanced by fixing these issues.

Reviewer #2 (Remarks to the Author):

Buchanan et al. present a new computer program called UnidecNMR for NMR peak picking. Their testing was performed on 1D to 4D NMR spectra. Authors claim UnidecNMR outperforms a number of freely available algorithms and produces similar quality of manually picked peaks.

Peak picking is a certainly one of the most challenging tasks in biomolecular NMR.

We would also say this is the case in chemical NMR.

However, it is unclear to me if UnidecNMR is a novel work and really a game changer because of a few things.

- As authors already wrote in the manuscript, there are already a few peak picking programs.

In the manuscript we demonstrate the performance of commonly encountered peak picking programs, and highlight deficiencies in their performance, establishing that there remains a need, in NMR, for a functional solution. We have included an additional challenging IDP and presented that the performance of methods freely available are simply not fit for purpose. In this case, more than 100 cross peaks in the HNCO and HNCA were missed. UnidecNMR gets them all.

- Using 2D peak list restraints sound like the restricted peak picking method which is already a common method in the field. Authors should clarify why it is different and novel.

We have included a comparison of sparky's restricted peak picking method with our own, and the performance of our own is far superior. This demonstrates that our method is different and novel. But more than that's, its performance makes it incredibly useful.

- What is reflection symmetry? Description is slim. It sounds like just seeing presence of a signal in the symmetric position.

We have clarified this in the text.

- Methods represented in here is very similar to ChiFit and Newton introduced decades ago (Chylla et al.). They read NMRPipe processing parameters for time-domain data simulation and fitting in addition to peak shape fitting on the frequency-domain data. FuDa and PINT are well-known peak shape fitting programs, and they have not been discussed and benchmarked while the UnidecNMR does peak shape fitting for deconvolution.

We apologise for not explaining this more clearly in the manuscript. Peak fitting is an entirely separate problem to peak identification. The software packages mentioned here (ChiFit, Newton, FuDa, PINT) fit peaks, following a user saying that there is a peak there. They are excellent packages. But before these packages even start, a user needs to identify where the peaks are. So these are not programs that help with 'peak picking'.

A least squares (or similar) optimisation is how these programs work. Our deconvolution algorithm is extremely different, working in an interactive fashion, converging on a final result as described. UnidecNMR is not in any way doing a peak shape fitting. We are effectively using a single peak shape to act as a 'filter' to analyse an entire spectrum, in fact.

Supporting only nmrPipe's file is also a limiting factor. Currently, many scientists cannot use nmrPipe due to technical difficulties and have moved to TopSpin, MNovo, etc for data processing. Therefore, nmrPipe is not always in their pipeline any more. When the peak picking is performed, it is very likely that users in the stage of using analysis programs like Sparky/Poky/CARA/CCPN/NMRView/ETC. Authors should be aware of that.

As discussed earlier in our response, our software takes raw FIDs straight from Bruker, and allows a user to process them prior to peak picking. All users will have the raw FIDs coming from either a

Bruker or legacy Varian spectrometer, and these are immediately compatible with our pipeline. We note the existence of free software such as Olivia that can easily convert between any of these formats.

Benchmark must illustrate more challenging problems. For example, data on figure 2 will be analyzed by a simple local maximum method easier, faster and more accurate not to mention analysis programs all have that algorithm alongside user-friendly interactive user interfaces for performing and improving by putting users' insights. Such as 3D C-13-NOESY and 2D-CC from a protein will be good candidates.

The data we have in the benchmark is not well analysed by a simple local maximum method. The SPARKY peak picker is basically this, and we would like to draw attention to the reviewer that its performance on our benchmark really is not good. This is particularly clear in the new 3D DDX4 dataset. If we could just take maxima, then there would be no computational or any difficult to solve when peak picking NMR data and the whole problem could be trivially solved. This is unfortunately not the case.

Reviewer #3 (Remarks to the Author):

The manuscript describes a peak picker which relies on deconvolution of peaks. This is a very established approach and does not have any novelty.

We would like to emphasise that deconvolution is not a single operation. There is no one 'deconvolution' algorithm or approach, and so we would respectfully disagree with the reviewer's generalisations here. Our specific unidec deconvolution algorithm was new in 2016, when we implemented this to analyse native mass spectrometry data, which has some similarities, but is not identical to for example the Richardson-Lucy algorithm. We cite the relevant literature which the interested party can follow.

Unidec is a new deconvolution algorithm that is now a standard in native mass spectrometry data analysis. This manuscript is our efforts to do the same for NMR. As discussed in the text, the NMR problem ends up quite a bit harder and so significant differences exist between Unidec for MS, and Unidec for NMR, specifically, the need for clustering. So we certainly do not agree with the reviewer in their assessment here that there is no novelty here, algorithmically speaking.

More than that, if it was the case that 'deconvolution' provides a generalised solution to peak picking, then it would be in widespread use by the community already. If the reviewer can point to a software package that outperforms ours in NMR data in 1-4D data, then we would most certainly agree with them that there is no novelty.

Yet, the authors implement it in a fashion that outperforms other algorithms and also even experienced spectroscopists according to the presented results.

We thank the reviewer and we agree. Fundamentally, the NMR community want a high performing peak picker that performs as well as a skilled user. This is what we are introducing.

I would like to state that it is impossible for a reviewer to judge these claims and in other fields the performance of algorithms are judged in competitions for example regarding structure prediction and so on. An algorithm which is not published and available cannot be subjected to such a test by the community. The results look plausible to me and therefore the manuscript and the algorithm should be published.

We also agree that the best way to assess performance would be through blind competition. We note in the letter that Prof. Jeff Hoch and his BMRB team have announced a blind peak picking contest and we have enthusiastically entered. But prior to this, there has been no such competition.

Still there is need for clarification and better description in parts. In Fig. 2C there is a comparison of the various algorithms regarding signal to noise and peak separation. It appears that peaks 1,4,7 are at the limit for reliably being picked by PICKY while just still reliably picked by UNIDEC. How do peaks look like that UNIDEC does not pick correctly? They should be included in Fig. 2 to give a feeling to the experienced spectroscopist where UNIDEC has its limits.

We have extensive demonstrations of cases where the different algorithms succeed and fail in the supplementary information.

The peak shape g that the user needs to provide is conveniently derived how?

We have expanded the description in the text and in the usage notes:

The next step is to obtain general peak shape parameters for use as the peak shape filter. There are a range of ways to do this. Firstly, a peak shape can be 'guessed', a trial UnidecNMR calculation is performed, the result inspected, then the values adjusted based on whether the result has obvious over or under picked the spectrum. Secondly, the UnidecNMR GUI loads in the most intense peaks detected, allowing a user to either manually or algorithmically fit them within the software (the 'Fit Peaks' tab). Sliders allow the various parameters to be adjusted until the shape is as desired.

In brief, the algorithm is reasonably tolerant to getting this wrong (Fig S1), but it helps to have a shape picked that makes things look basically correct when comparing simulated and raw data. The GUI has routines to help fit a clutch of intense peaks together, to find values to parametrise this function as a first guess.

There are two empirical factors *fac* and *squash* which modify the peak shape and optimized values are given for them: (*fac*=1.6, *squash*=0.625) Are these values universal? Or do they need to be adjusted for each spectrum? If (and this is my impression from the manuscript) they are universal, the authors might provide some insight why these values are optimal and how that could be explained given the mathematical background of the procedure. The authors write: "Below these thresholds the success of the algorithm falls away from 100%, and its success depends on the exact shape of the noise profile in an individual spectrum. It nevertheless remains reasonably successful outside these windows, and typically fails in cases where an experienced user would also be unconfident." It would be great if the authors could provide for example in the SI such examples that the experienced spectroscopist can convince himself of the signal quality under these circumstances.

We have expanded discussion on this. The algorithm itself has two fudge numbers, the 'fac' and the 'squash'. On synthetic data, we determine that there is a plateau of values where performance is excellent. In the distribution we have fixed 'squash' to 0.75 of the FWHM of the peak shape used, meaning that the user has just one number to play with, 'fac'. In the paper, we have effectively used two values, 1.4 on 2D and 1D data, and 1.6 for 3D and 4D. We envisage that a user will play with this a little bit on a case-by-case basis but the defaults are basically very good, in our hands.

In Fig. 4 there are comparisons on 2D spectra and UNIDEC outperforms all comparisons. Specifically the number of wrongly identified peaks is much smaller in UNIDEC than in the other programs. Still UNIDEC did wrong peaks and it would be interesting to see what the features of these peaks are.

We have a detailed discussion on these points in the supplementary figures. For the newly added DDX4 dataset we have a detailed comparison where unidecNMR gets it wrong compared to a user. These are all on the edges of what a human would pick if analysing these spectra in isolation, but we have classified these as errors in order to be as critical as possible.

Would they be identified by an assignment program working with peak lists such as MARS or FLYA because some spurious noise was picked but the peaks would not lead to wrong assignments?

These programs, to our understanding are working on assimilating data from multiple spectra. This would be downstream of what we are accomplishing here. So for example, sending our peak lists to the MARS program and similar would be an excellent next step for the purpose of biomolecular assignment. Testing this is outside the scope of this manuscript but is an exciting next step and an obvious place for us to go next.

Regarding the discussion of Fig. 3, is it right to assume that the 1D peak picking of multiplets does identify i.e. a doublet of doublets as such but just identifies 4 lines? Are the peak positions so accurate that one could assemble e.g. a doublet of doublets from the fact that frequency difference of peak 1 and 2 should be the same as 3 and 4; and that frequency difference of peak 1 and 3 should be the same as 2 and 4, maybe taking the intensity (or integral which is available since the peak shape is defined) into account additionally? Along the same lines, would it be obvious from integrals which peaks belong to the same molecule in case of mixtures with different concentrations?

Downstream it is perfectly feasible to have additional heuristics to look for patterns to do these types of functions. This very interesting problem is outside the scope of this manuscript.

Regarding the statement: "One spectrum of note was an HNC0 acquired on Hsp16.5 (unpublished data), where the decoupler was mis-set, resulting in a triplet for each peak in the ¹³C dimension (Fig. S1b). By increasing the effective peak width used for calculations, we were able to account for this deficiency. By contrast, the other algorithms found analysing this specific experiment highly challenging." The authors should answer the following questions: In the 1D case, multiplets were

assigned. Why could such an approach not be used to here also given the fact that the couplings are known: 7 and 11 Hz is the splitting which could be incorporated in the peak shape

We can also set the peak shape to something more appropriate for the singles here and in this case the algorithm picks the 3 parts of the 'triplet'. If this spectra were used for the purpose of biomolecular resonance assignment, a user would want to know the average, and would not be that interested in the artefacts.

g. The remark "this specific experiment highly challenging" needs a more scientific explanation.

We have rephrased this.

In the case of the methyl NOESY, 60% more peaks were found than by an experienced user. It would be great to see some of those additional peaks in the spectrum which were missed by an experienced user. Fig. 6 does not really report this information. Is it because of signal to noise or because of overlap of resonances? Some peaks NOESY peaks refer to distances a little larger than 14 Angstrom. For loops whose dynamics was not included in the pdb file of reference this makes sense. Do all those peaks with long distances belong to the loop? 10 Angstrom is quite a long distance. What mixing time was used in that NOESY? In addition, in Fig. 6iii there are quite a number of peaks with relative cross peak intensity of 0. How can these peaks be found if their relative and then necessarily absolute intensity is 0/close to 0?

In a deuterated methyl labelled sample, the proton density is very low when compared to a uniformly protonated sample. As a result, NOEs go a lot further in these types of samples. The 14A NOEs map to areas within the protein that have low methyl density.

The specific cases here are described in detail in this paper, and are cases where the proteins are rigid with a good reference structure. Certainly the highest intensity NOEs have to come from relatively short distances, but conformation dynamics and orientational effects can lead to short distances giving both intense and not intense cross peaks.

It is very likely, in our view, that in the detail here it is possible to obtain a richer picture of the conformational dynamics but there is no known current method that can extract this. For the present work, it is important to note, as we have done, that the NOEs identified are 'sensible' in the sense that they are compatible with the known structure in the context of the measurement.

We invite the reviewer to start here for a discussion on the subject:
<https://pubs.acs.org/doi/abs/10.1021/jacs.6b11358>

The ERC grant number is not stated in the acknowledgment.

Fixed!

Reviewer response.

We were delighted to see positive responses from reviewers 1 and 3.

Reviewer 1:

"I think the authors have done a very thorough job of answering this reviewers comments and am happy with paper in its latest form and would recommend publication"

"The software was installed and tested and worked as described and the demo videos were accurate and useable."

Reviewer 3:

"all my comments are taken care of adequately."

Reviewer 2 expressed concerns that they were unable to download and test our software. We supplied google drive links to the code, which the reviewer appears to have used and referenced in their response to us and so we are not entirely sure how to further address their concerns.

We envisage this to be an easy-to-use package for the NMR community. To this end for this revision, similar to what was previously provided, the editor will send to you.

1. The code in a form that a user can use and install on either mac, windows or linux.
2. A link to a youtube instructional video to showing people how to use it.
3. The complete benchmark containing 2D HSQC and 3D HNCO/HNCA data used for benchmarking as described in the manuscript.

With the kind support of Prof Jeff Hoch and his team at NMRbox, the software and data is available now on the NMRBox platform. Below we provide instructions for the reviewers to anonymously log in to nmrBox to test if they cannot install the software locally.

The code is currently not available on www.unidec.chem.ox.ac.uk. But it will be once the article has been accepted for publication. We would like the home for this work to be nature communications.

Following discussions with the editors, as described in the manuscript, the core binary itself is written in C++. We will distribute this as a pre-compiled binary. The GUI, containing the routines that calls the binary are written in python, and so are entirely adaptable and human readable by people that like python. The C++ code itself will not be shared following instruction by Oxford University's commercial arm. However, all steps necessary to re-create this code should any scientist programmer wish to, are provided in the manuscript. The code algorithm itself is actually very simple and described in the original paper where we developed it for mass spectrometry use. The single equation at it score is stated in the paper with the following citation:

<https://pubs.acs.org/doi/abs/10.1021/acs.analchem.5b00140>

And adaptions necessary to make this work with NMR data are in the manuscript.

We have prepared our benchmark of raw data and set it up in a manner that allows a user to run the entire calculation starting at raw FIDs to get processed and peak-picked spectra for the 12 spectra in the benchmark with a single command. A user can do this locally following our installation instructions, or on nmrBox if installation is either arduous or unwelcome.

Note that the benchmark includes a range of difficulty, both large and small proteins, with and without exchange broadening, and an IDP. There is both varian and Bruker data formats, and both uniformly and non-uniformly sampled data as described in the manuscript. It forms a good test for data of this type and going forward we will welcome contributions from the community.

The community would be well served by a on-going database of this type. We believe that we are the first to make on that is truly open access, setup in a way to easily perform comparisons of the success of peak picking and we hope going forward that either other users contribute to this, or someone else takes our data and makes something better.

INSTRUCTIONS FOR REVIEWERS TO USE NMRBOX ANONYMOUSLY:

To help you test the UnidecNMR algorithm, we have installed the software on an NMRBox virtual machine. To keep your identities anonymous, the NMRBox team have set up a reviewer account. To use this, please perform the following tasks:

- 1) Download and install RealVNC from the following link: <https://www.realvnc.com/en/connect/download/>
- 2) Once installed, we can open VNC Viewer and create a new connection (example screenshots are shown for Mac)

- 3) Please enter unidec.nmrbox.org as the VNC server

4) Once you have pressed OK, a new icon for UnidecNMR reviewer NMRBox will appear in the VNC Viewer window. Double click to connect and use the following anonymous credentials:

Username: [REDACTED]
Password: [REDACTED]

[REDACTED]

5) once you press ok you should have successfully logged into the NMRBox, where you can open a terminal window and perform the following tests:

[REDACTED]

Example UnidecNMR window

change to the example directory in the home folder by typing:

```
cd ~/example
```

Then

```
unidecNMR
```

This will spawn a UnidecNMR window. We then recommend following our YouTube videos for more instructions.

Running the UnidecNMR Benchmark

To run the UnidecNMR benchmark, please navigate to the ~/UnidecNMR_data by typing

```
cd ~/UnidecNMR_data
```

Then

```
unidecBenchmark
```

The benchmark should now start running, and will complete full processing and peak picking of the spectra using the settings contained within each deconParFile. The results will be outputted to the screen, and to a benchUnidec.YYYY-MM-DD.log file.

Spectra and results for each component of the dataset can also be analysed by navigating through the folders and spawning a unidecNMR instance. For example, you can view the alphaB HNC0 results by changing directory (using the cd command) to ~/UnidecNMR_data/alphaB/hnc0 and then typing unidecNMR. Further instructions for using UnidecNMR window can be found on our YouTube tutorials.

In the following we respond to specific issues raised by reviewer 2 following the last submission.

Reviewer #2 (Remarks to the Author):

[response from us in BLUE, reviewer in BLACK]

1. The provided link for UnidecNMR in the manuscript does not work.

In the paper we have a link to unidec.chem.ox.ac.uk. This link will go live on publication. For the purpose of review, we supplied the reviewers with the software and example data for testing. We note reviewer 1 has previously commented:

"The software was installed and tested and worked as described and the demo videos were accurate and useable."

2. "Nature Comm." is open access. Is the UnidecNMR itself open source? It appears that only the wrapper to call the program is open source from the given code.

We have discussed the software policy with the journal and our method of distribution where we have the python source code (including the GUI) with a pre-compiled C++ binaries is acceptable with their distribution policy. The python calls the pre-compiled binary, but a user can call the binary directly from the command line.

The authors have provided a compiled version, so I cannot evaluate it.

This is a challenging statement for us to parse. Clearly the reviewer has downloaded our software and so we do not see why they cannot evaluate it as the other reviewers were able.

Additionally, the HNCO and HNCA examples, claiming superiority, haven't been provided. This is necessary because the "peak-picking" process can be biased by different parameters like contour settings.

In the original download, the reviewer might see a folder marked 'EXAMPLES'. We provided this precisely for the reviewers to acquaint themselves with our software and platform. We also provided a youtube video to show users how they can do this.

In the manuscript we stated that we will release our complete benchmark of raw data. We have prepared a link as described above containing more of the raw data. This will be distributed with the software to serve as a community resource going forward as described above.

3. Is this a decomposition or deconvolution? This work seems to be neither of those but rather filtering. Please elaborate.

We are performing a deconvolution. We have noted this in the original manuscript and in our specific responses to reviewer 2 in our revision letter. A subtle and perhaps confusing point is that the algorithm can be used as a resolution enhancing filter, as shown in the manuscript.

Deconvolution is a phrase often used colloquially and imprecisely in the NMR literature. The specific procedure we are doing is iterative, that should not be confused with 'fitting' or an optimisation, the

types of calculations the field typically performs. It should also be understood that there is no single 'deconvolution' algorithm. For edification, a good place to start is the following wiki page:

https://en.wikipedia.org/wiki/Richardson–Lucy_deconvolution

Our algorithm is similar to, but not identical to this important iterative 1972 algorithm. A complete discussion on the relationship between this algorithm and our unidex algorithm (originally designed for native MS data) can be found within the original unidex paper (that we cite, as described above).

<https://pubs.acs.org/doi/abs/10.1021/acs.analchem.5b00140>

We have provided all the information necessary to an interested person to understand the algorithm and have provided the single key iterative equation in the manuscript. Any interested reader has a path necessary from citations we provide to find out more, as is convention for the literature.

We would welcome a constructive discussion on the finer points of specific algorithms and their evolution but for the purpose of this manuscript, we believe in the manuscript we have already provided both the top level summary and the necessary links to allow a deeper dive for the curious.

4. Taking raw FIDs directly can be a drawback. NMRPipe, TopSpin, and NMRFX Processor programs have sufficient functionalities to improve the quality of the data, and users will not be able to benefit from those programs in their pipelines.

We respectfully disagree with the reviewer. Once a user has a spectrum, created using their favourite piece of software, all common NMR formats can be easily interconverted. A non-exhaustive list for ways to do this: 1. using python (for example going via the excellent and free nmrplug module), 2. using tools available on nmrbox, 3. through using other freely downloadable software like Olivia. The two cases noted, TopSpin and NMRFX. are easily converted to nmrPipe by these methods. You do not need an nmrPipe license or the software installed to use this format.

This means we are in no way restricting a user to picking one processing format over another when we make our program take nmrPipe files as the input. A user can take whatever spectrum they have, and convert into an nmrPipe format.

If a user doesn't have any strong opinions about their preferred processing software, they can take raw FIDs straight from the spectrometer (Bruker or Agilent), and do all processing they would like to do within our software. Commonly processing functions including apodization, zero filling, linear prediction and non-uniform spectral reconstruction using the SMILE engine are then all easily available using nmrPipe and to our knowledge is the most widely used, and so this is why we focus on this.

It would be a straightforward exercise to set our python GUI up to use any other preferred processing software. As part of this project we considered do the entire thing ourselves. The only thing that stopped us was non-uniform data reconstruction, which remains itself a challenging problem for which there are several solutions available (including MDD embedded inside modern versions of Topspin).

But to stress again, we are not restricting user choice. We providing a tool that can be added into any NMR workflow. If a user wishes to process their data in their preferred format, they can convert it post-processing to the nmrPipe format and it will work with our software.

We are unaware of any function to improve data 'quality' that cannot be performed by the nmrPipe intrinsic processing functions. All the functions that our group, and to our knowledge both the

chemical and biological NMR communities use routinely for data analysis are readily available within nmrPipe. This can be tricky thing to learn and use, which is why we allow users to access these functions with a few clicks using our software and GUI.

5. Again, the authors insist on UnidecNMR's great performance. Please provide specific instructions or examples for how I can test UniDecNMR's performance.

Data, software and a youtube video were provided as part of the reviewing process to allow the reviewers to do this. Reviewers 1 and 3 were able to do this with the information provided. Please see the above letter for how to access code and the benchmark data, as well as instructions for how to access this via nmrBox.

We have additionally now provided the HNCO/HNCA/HSQC data for Ddx4, Ubiquitin, Hsp27 and alphaB-crystallin as described above. We believe we have provided a tool that the NMR community can embrace and adapt to their own needs.